# Pathway-specific dysregulation of striatal excitatory synapses by LRRK2 mutations

**Chuyu Chen[1], Giulia Soto[2], Vasin Dumrongprechachan[2], Nicholas Bannon[2], Shuo Kang[1], Yevgenia Kozorovitskiy[2]\*, Loukia Parisiadou[1]\***

[1]Department of Pharmacology, Feinberg School of Medicine, Northwestern University, Chicago, United States; [2]Department of Neurobiology, Northwestern University, Chicago, United States

**Abstract** LRRK2 is a kinase expressed in striatal spiny projection neurons (SPNs), cells which lose dopaminergic input in Parkinson's disease (PD). R1441C and G2019S are the most common pathogenic mutations of LRRK. How these mutations alter the structure and function of individual synapses on direct and indirect pathway SPNs is unknown and may reveal pre-clinical changes in dopamine-recipient neurons that predispose toward disease. Here, R1441C and G2019S knock-in mice enabled thorough evaluation of dendritic spines and synapses on pathway-identified SPNs. Biochemical synaptic preparations and super-resolution imaging revealed increased levels and altered organization of glutamatergic AMPA receptors in LRRK2 mutants. Relatedly, decreased frequency of miniature excitatory post-synaptic currents accompanied changes in dendritic spine nano-architecture, and single-synapse currents, evaluated using two-photon glutamate uncaging. Overall, LRRK2 mutations reshaped synaptic structure and function, an effect exaggerated in R1441C dSPNs. These data open the possibility of new neuroprotective therapies aimed at SPN synapse function, prior to disease onset.

**\*For correspondence:**
Yevgenia.Kozorovitskiy@
northwestern.edu (YK);
loukia.parisiadou@northwestern.
edu (LP)

**Competing interests:** The authors declare that no competing interests exist.

## Introduction

Gain-of-function mutations in the leucine-rich repeat kinase 2 (*LRRK2*) gene represent the most common cause of familial Parkinson's disease (PD) (*Alessi and Sammler, 2018*). Carriers are at risk for late onset PD, which is clinically indistinguishable from sporadic PD, consistent with a possibility of common disease mechanisms (*Kluss et al., 2019*; *Italian Parkinson's Genetics Network et al., 2006*; *Haugarvoll et al., 2008*; *Di Maio et al., 2018*). *LRRK2* gene product is a large multi-domain protein with two catalytic domains: a GTPase (ROC-COR) domain and a serine/threonine-directed protein kinase domain. Pathogenic mutations are found predominantly in these two domains, suggesting that LRRK2 enzymatic activities are involved in PD pathogenesis (*Cookson, 2010*), (*Esteves et al., 2014*). Yet, how LRRK2 mutations in the two distinct functional domains contribute to PD pathogenesis and whether they act through a common mechanism is unknown. Enhanced LRRK2 kinase activity conferred by the G2019S (GS) mutation in the kinase domain is the most extensively studied property of mutant LRRK2 (*Cookson, 2010*), (*Steger et al., 2016*). Meanwhile, the R1441C (RC) substitution in the GTPase domain results in impaired GTP hydrolysis, which is thought to indirectly enhance kinase activity through mechanisms that remain to be determined (*Nguyen and Moore, 2017*), (*Xiong et al., 2010*).

Despite remaining questions, the last decade marks extensive progress in our understanding of LRRK2 function. This kinase is highly expressed in the spiny projection neurons (SPNs) of the striatum (*Nguyen and Moore, 2017*; *West et al., 2014*; *Parisiadou et al., 2014*). LRRK2 expression peaks in a developmental time window of extensive glutamatergic excitatory synapse formation, suggesting that LRRK2 may regulate the development or function of excitatory synaptic networks. Consistently, several lines of evidence suggest that loss of LRRK2 alters striatal circuits during postnatal

**eLife digest** Parkinson's disease is caused by progressive damage to regions of the brain that regulate movement. This leads to a loss in nerve cells that produce a signaling molecule called dopamine, and causes patients to experience shakiness, slow movement and stiffness. When dopamine is released, it travels to a part of the brain known as the striatum, where it is received by cells called spiny projection neurons (SPNs), which are rich in a protein called LRRK2. Mutations in this protein have been shown to cause the motor impairments associated with Parkinson's disease.

SPNs send signals to other regions of the brain either via a 'direct' route, which promotes movement, or an 'indirect' route, which suppresses movement. Previous studies suggest that mutations in the gene for LRRK2 influence the activity of these pathways even before dopamine signaling has been lost. Yet, it remained unclear how different mutations independently affected each pathway. To investigate this further, Chen et al. studied two of the mutations most commonly found in the human gene for LRRK2, known as G2019S and R1441C. This involved introducing one of these mutations in to the genetic code of mice, and using fluorescent proteins to mark single SPNs in either the direct or indirect pathway.

The experiments showed that both mutations disrupted the connections between SPNs in the direct and indirect pathway, which altered the activity of nerve cells in the striatum. Chen et al. found that individual connections were more strongly affected by the R1441C mutation. Further experiments showed that this was caused by the re-organization of a receptor protein in the nerve cells of the direct pathway, which increased how SPNs responded to inputs from other nerve cells.

These findings suggest that LRRK2 mutations disrupt neural activity in the striatum before dopamine levels become depleted. This discovery could help researchers identify new therapies for treating the early stages of Parkinson's disease before the symptoms of dopamine loss arise.

development (*Parisiadou et al., 2014*), and the GS pathogenic mutation increases glutamatergic activity in cultured cortical neurons (*Beccano-Kelly et al., 2015*) as well as in acute striatal slices (*Matikainen-Ankney et al., 2016*), (*Volta et al., 2017*). Recent studies assign a critical role of LRRK2 in presynaptic terminal vesicle function (*Pan et al., 2017*). Here, the GS mutation impairs presynaptic glutamatergic release, suggested to underlie changes in glutamatergic activity of striatal neurons (*Volta et al., 2017*). The potential postsynaptic function of LRRK2 remains less well-characterized. We have previously shown that the RC mutation impedes normal striatal protein kinase A (PKA) signaling, which in turn results in increased GluA1 phosphorylation in developing SPNs, consistent with a postsynaptic mechanism of action (*Parisiadou et al., 2014*). Similarly, glutamate receptor trafficking perturbations were observed in *GS* knock-in (KI) mice in response to plasticity induction protocols (*Matikainen-Ankney et al., 2018*). Furthermore, although an increase in spontaneous excitatory postsynaptic currents has been reported for the dorsomedial striatum, a recent report failed to show this phenotype for the ventral striatum (*Huntley, 2020*), despite LRRK2 expression in that region (*West et al., 2014*), (*Giesert et al., 2013*). These observations suggest that LRRK2 mutations may shape the corticostriatal synaptic function in a synapse-, cell- and area-specific manner. Overall, despite the links between LRRK2 and glutamatergic synapse dysfunction, the field currently lacks a coherent framework for understanding how the two distinct LRRK2 mutations selectively alter synaptic function in specific cell types.

Given the complementary role of direct and indirect striatal pathways in behavior (*Kravitz et al., 2010*; *Fieblinger et al., 2014*; *Kozorovitskiy et al., 2012*) the lack of pathway specificity in prior studies limits our understanding of disease related mechanisms associated with LRRK2 mutations. Previous studies did not compare the dysfunction of two LRRK2 mutations which are found in distinct LRRK2 domains and confer divergent biochemical properties to the kinase, choosing instead to focus on the one mutation or the other. Earlier reports have mainly focused on the GS pathogenic mutation, and whether the molecular mechanisms underlying pathology across the two most common mutations remain unknown. In addition, the phenotype of LRRK2 mutant models is subtle and corresponds to a moderate susceptibility for PD. Therefore, refined tools are required to distinguish the early pre-pathology functional abnormalities. We have taken the approach of combining molecular, anatomical and electrophysiological approaches that capture global aspects of LRRK2 function in the

striatum, but also single synapse-specific effects, in case abnormalities are associated with specific synapse subtypes.

Here, we undertook a systematic synapse function and structure analysis in two different mutant LRRK2 mouse lines to evaluate the contribution of this kinase mutations to SPN synapses across direct and indirect pathways. Using a combination of subcellular fractionation biochemistry, super-resolution imaging, and two-photon laser scanning microscopy with whole-cell physiology approaches, we found a critical role for LRRK2 RC, and to a lesser extent GS mutation, in organizing the structure and function of the SPN excitatory synapses, particularly for dSPNs.

## Results

### LRRK2 +/RC mutation leads to increased synaptic incorporation of GluA1 in the striatum

In order to resolve LRRK2 mediated synaptic alterations in identified SPNs in a systematic way, we crossed the pathway-specific reporter BAC transgenic mice (*Tozzi et al., 2018a*) with two mutant LRRK2 KI mouse lines. Specifically, *Drd1-Tomato* and *Drd2-eGFP* reporter mice were crossed with each of the two mutant LRRK2 lines (RC and GS) (*Figure 1A*). This platform enabled pathway-specific interrogation of each LRRK2 mutation side by side, as well as initial experiments described below that were done on aggregated SPN populations. Our previous findings showed that *RC* KI neurons displayed altered PKARIIβ localization, as compared to wild-type (+/+) neurons, suggesting altered synaptic PKA activity in SPNs (*Parisiadou et al., 2014*). To further explore the synaptic PKA signaling in both KI mouse lines, we employed differential centrifugation, discontinuous sucrose gradient, and detergent extractions to isolate synaptic subcellular fractions from striatal extracts as shown in *Figure 1B* (*Parisiadou et al., 2014*), (*Bermejo et al., 2014*; *Peng et al., 2004*). Consistent with our previous data, PKA activity was found elevated in the P2 crude synaptosomal preparation of *+/RC* but not *+/GS* striatal extracts, when phospho-PKA substrate antibody was used to detect phosphorylation of downstream PKA targets. The increased PKA activity in *+/RC* was further confirmed by an increase in phosphorylation of GluA1, a key downstream target of PKA. To directly evaluate LRRK2 function, we examined the phosphorylation of two known targets of this kinase, Rab8A and Rab10. Phosphorylation of both targets was selectively increased in the synaptic fraction (P2) of the RC striatal extracts (*Figure 1D*). Notably, while Rab8A is a known physiological target of LRRK2 kinase activity (*Steger et al., 2016*), (*Bonet-Ponce and Cookson, 2019*), it was unclear whether this is the case in the striatum, since only low levels of *Rab8A* transcripts are found in dSPNs and iSPNs at the single-cell level (dropviz.org) (*Saunders et al., 2018*). Our finding stands in accordance with two recent studies in cell lines, showing that the RC and other mutations found in the GTPase domain of LRRK2 result in greater increase kinase activity, compared to the GS mutation (*Purlyte et al., 2018*; *Liu et al., 2018*). We further isolated the post-synaptic density (PSD) fraction from the P2 preparation by using a detergent extraction step. The PSD fraction showed enrichment of the postsynaptic marker PSD95, but not presynaptic protein synaptophysin (svp38) (*Figure 1C*). Given that PKA mediated phosphorylation of S845 in GluA1 has a significant impact on synaptic trafficking of GluA1 (*Parisiadou et al., 2014*), (*Roche et al., 1996*), we examined the levels of GluA1 in the PSD fractions of KI mice. We observed a significant increase in the GluA1 levels in the PSD fraction from *+/RC*, but not *+/GS* mice, compared to wild-type control (*+/RC*: 47% increase of control; *+/GS*: 19% increase relative to control, one-way ANOVA p=0.0397, post-hoc tests as noted, n = 6 per group) (*Figure 1E,F*), which was paralleled by elevated PKA activity specifically in the PSD fractions in *+/RC* (*+/RC*, 167% of control, one-way ANOVA p=0.0486, post-hoc tests as noted, n = 6) but not *+/GS* mice (*+/GS*, 118% of control, n = 6) (*Figure 1E,G*). Next, we determined the relative levels of PSD-95 protein in the PSD fraction across genotypes, which was used in the ratios in *Figure 1F,G*. We run subcellular fractions of all genotypes on the same blot (*Figure 1—figure supplement 1A,B*), and normalized PSD95 to another synaptic protein, Homer1. PSD95 band intensities across fractions and genotypes showed no difference. Furthermore, PSD95 band intensities in each of S1, P2, and PSD fractions were expressed as percentage of the PSD95 intensity of the wild type in each fraction (*Figure 1—figure supplement 1C*). Similarly, no differences in PSD95 levels in mutant LRRK2 fractions were observed after one-way ANOVA. Overall, our data show increased GluA1 levels in the PSD fractions of *+/RC* striatal extracts, which parallels enhanced synaptic PKA signaling.

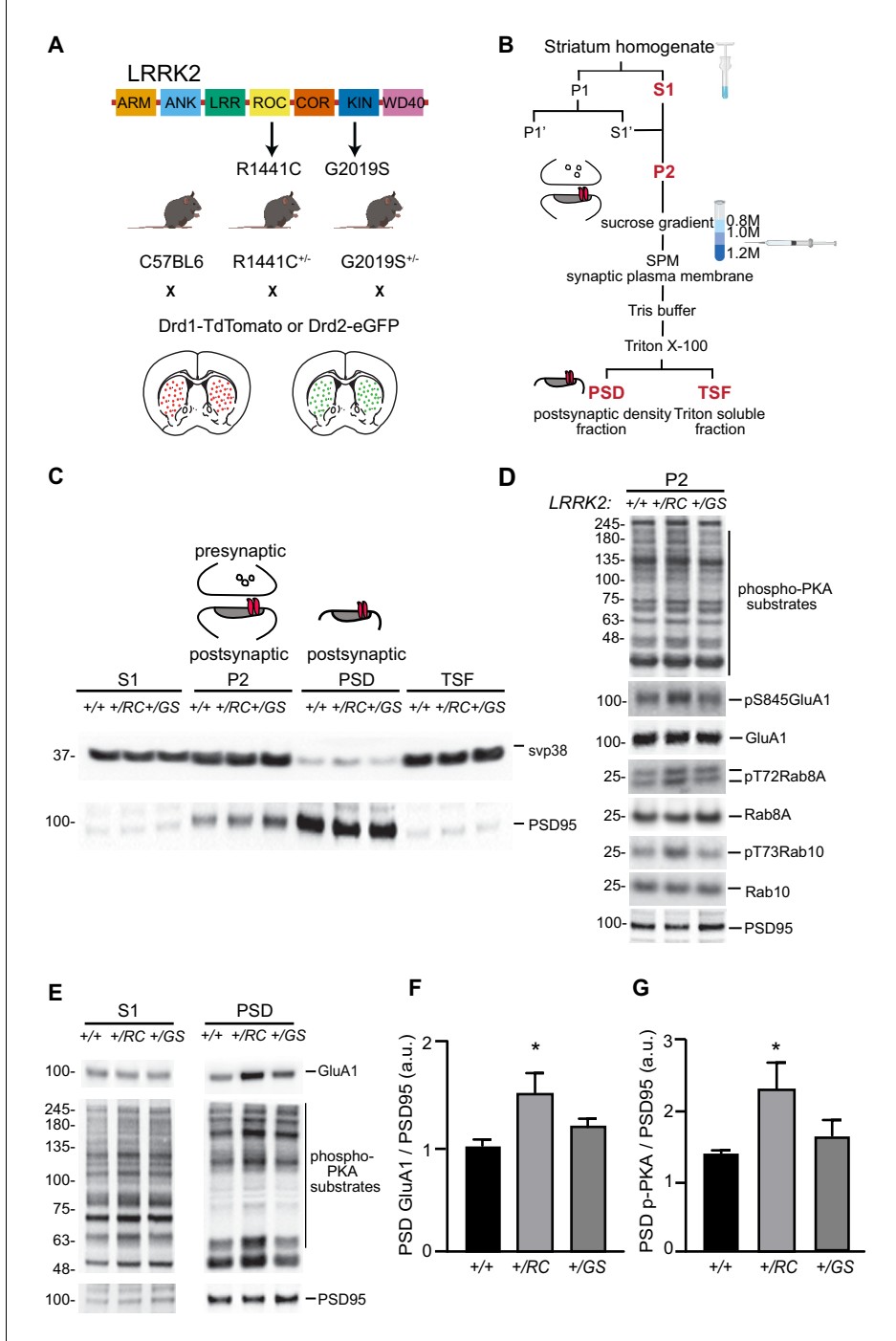

**Figure 1.** LRRK2 RC mutation increases synaptic glutamate receptor content in the striatum. (**A**) Schematic diagram of LRRK2 protein highlighting the armadillo repeats (ARM), ankyrin (ANK) repeats, Ras of complex (ROC), C-terminal of ROC (COR), kin (KIN), and WD40 domains. Knock- in mice expressing the R1441C and G2019S mutations found in the ROC and kinase domains respectively, crossed with either *Drd1-Tomato* or *Drd2-eGFP* mouse lines. (**B**) Workflow schematic for subcellular fractionation of striatal homogenate for the enrichment of postsynaptic density fraction (PSD). (**C**) Representative western blot analysis of the subcellular fractionation results, showing supernatant (S1), crude synaptosomal preparation (P2), PSD, and Triton soluble fractions (TSF). (**D**) Western blot analysis of *+/+, +/RC,* and *+/GS* P2 striatal fractions probed for p-PKA substrates, pS845 GluA1, total GluA1, pT72Rab8A, total Rab8A, pT73Rab10, total Rab10, and PSD95. (E) Western blot analysis of *+/+, +/RC,* and *+/GS* mice probed for GluA1, p-PKA, and PSD95. S1 and PSD fractions are shown. (**F-G**) Quantification of GluA1
*Figure 1 continued on next page*

*Figure 1 continued*

and p-PKA proteins in PSD fractions normalized to PSD95. Summary graphs reflect the mean, error bars reflect SEM. *p<0.05, Tukey post-hoc test following one-way ANOVA.

The online version of this article includes the following source data and figure supplement(s) for figure 1:

**Source data 1.** Numerical data of the graphs in *Figure 1*.
**Figure supplement 1.** LRRk2 mutations do not alter PSD95 levels.
**Figure supplement 1—source data 1.** Numerical data of the graphs in *Figure 1—figure supplement 1*.

## Altered nanoscale organization in the dendritic spines of direct and indirect pathway mutant LRRK2 SPNs

To precisely examine the nanoscale organization of GluA1 receptors in the dendritic spines of identified SPNs in mutant LRRK2 mice, we used structured illumination microscopy (SIM). SIM is able to overcome the resolution limits of conventional microscopy, and recently revealed a high degree of organization among scaffold proteins and AMPA receptors, forming nanoscale subsynaptic domains (*Gao et al., 2018*; *Crosby et al., 2019*; *Smith et al., 2014*). Primary cultured striatal neurons from *Drd1-dTomato* or *Drd2-eGFP* mice crossed with either *+/RC* or *+/GS* mouse lines (*Figure 2A*), were immunostained for GluA1 and PSD95 (*Figure 2B*, and *Figure 3A–B*, *Figure 2—video 1*). We found several lines of evidence demonstrating increased synaptic incorporation of GluA1 in *+/RC* dSPNs, compared to *+/+* ones. Specifically, the distance between GluA1 and PSD95 was smaller in *+/RC* dSPNs compared to control dSPNs, while no difference was observed between *+/GS* and *+/+* neurons (+/RC, 63% of control, +/GS 85% of control one-way ANOVA, p=0.0208, post-hoc tests as noted, n = 75–93 dendritic spines/genotype) (*Figure 2C–D*). Similarly, the *+/RC* dendritic spines showed the greatest shift in cumulative distribution for minimum distance between GluA1 and PSD95 nanodomains (*Figure 2E*). While the overlap area of GluA1 and PSD95 nanodomains was found elevated in the dendritic spines of *+/RC* dSPNs, no differences were observed in *+/GS* neurons, compared to *+/+* neurons (+/RC, 137% of control, +/GS, 102% of control one-way ANOVA p=0.0215, post hoc tests as noted, n = 75–93 dendritic spines/genotype) (*Figure 2F*). Similarly, the cumulative distribution analysis showed that GluA1-PSD95 distance was shifted towards higher values in the *+/RC* dSPNs (*Figure 2G*).

Given that the PSD95 area in *+/GS* dSPNs showed a trend towards decrease (+/RC, 94% of control, +/GS, 83% of control, one-way ANOVA p=0.0904, post-hoc +/+ vs +/GS p=0.075) (*Figure 2—figure supplement 1A,B*), we evaluated the percentage of the GluA1-PSD95 overlap relative to the total PSD95 area across genotypes. 32% of total PSD95 area contained GluA1 in *+/RC* dSPNs, compared to 23% in *+/+* neurons, and 26% in *+/GS* neurons (one-way ANOVA p=0.0015, post hoc tests as noted) (*Figure 2H*). These data strongly suggest increased levels of GluA1 at the synapses of +/RC dSPNs. Correlation analysis was performed to investigate the association between GluA1-PSD95 overlap nanodomain area and PSD95 area. A positive correlation was observed in all three genotypes (control, Spearman r = 0.5822, p-value<0.0001, +/RC, r = 0.6522, p<0.0001 and +/GS r = 0.6569, p-value<0.0001). The correlation was stronger in LRRK2 mutant SPNs (*Figure 2I–K*).

A similar synaptic nanoscale organization analysis was performed for *Drd2-eGFP* control and mutant LRRK2 SPNs. The neurons were immunostained with GluA1 (purple), and PSD95 (orange) (*Figure 3A–B*) and minimum distance summary data (+/RC, 87% of control, +/GS, 89% of control, one-way ANOVA, p=0.6435 n = 72–88 dendritic spines/genotype), as well as cumulative distribution of the distance between the GluA1 and PSD95 nanodomains, showed no difference across genotypes (*Figure 3C–D,E*). We observed that the overlap GluA1 and PSD95 area was increased in *+/GS* SPNs (+/RC, 105% of control, +/GS, 155% of control, one-way ANOVA, p<0.0001; post hoc tests as noted, p=72–88 dendritic spines/genotype) (*Figure 3F*). Further analysis showed an increase in the mean PSD95 area in the *+/GS* neurons (+/RC, 123% of control, +/GS, 137% of control, one-way ANOVA, p=0.0008, post hoc tests as noted) (*Figure 2—figure supplement 1C*), as well as in cumulative frequency (*Figure 2—figure supplement 1D*). However, the percentage of the total PSD95 area containing GluA1 nanodomains *+/GS* iSPNs compared to controls was similar (one-way ANOVA, p=0.469) (*Figure 3H*). Thus, the increased size of GluA1-PSD95 overlapping domains in the *+/GS* iSPNs reflects the elevated PSD95 area of these neurons (*Figure 2—figure supplement 1C-D*), and not synaptic GluA1 incorporation, as found in *+/RC*.

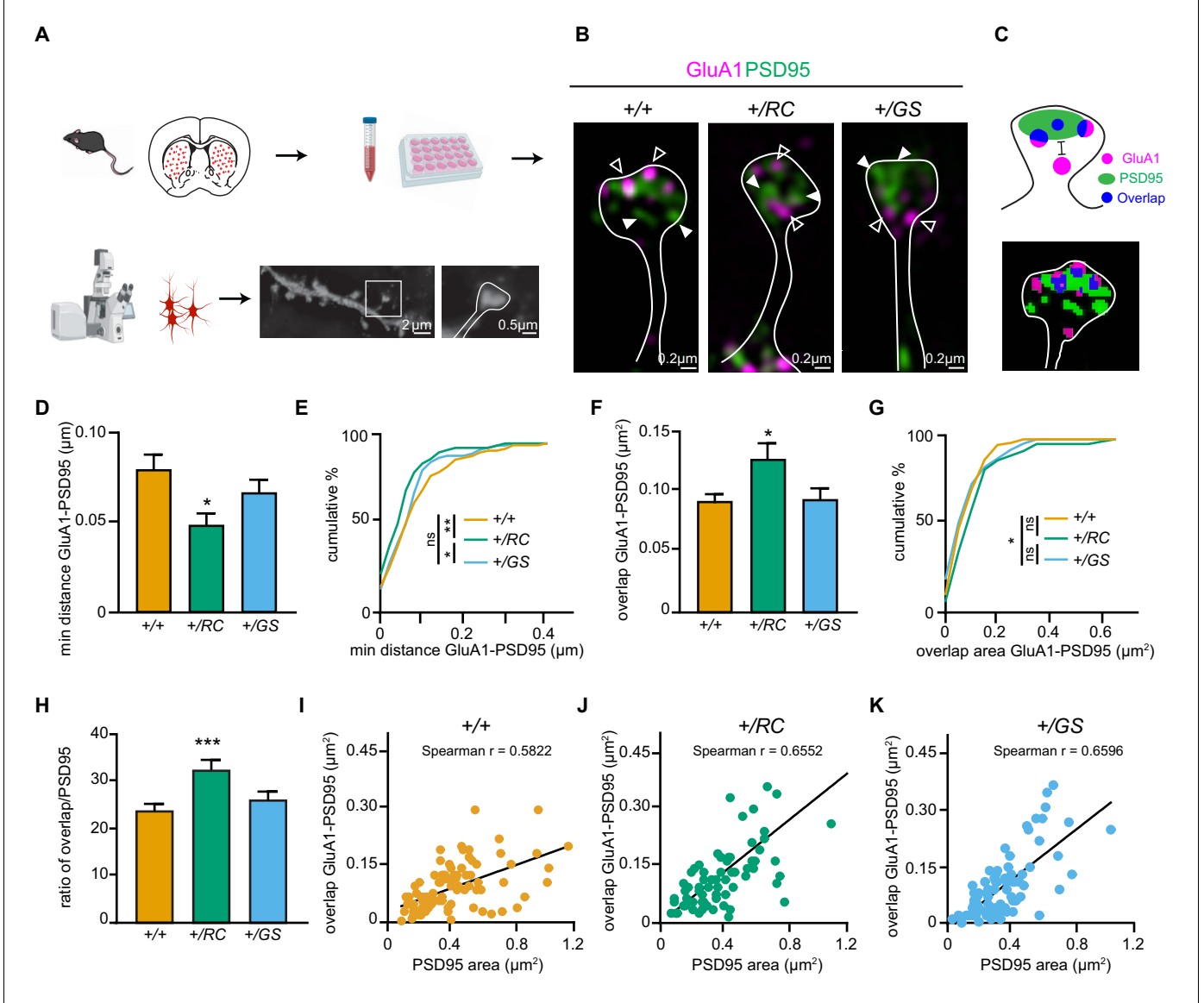

**Figure 2.** LRRK2 RC mutation restructures the nanoscale synaptic organization of dSPNs. (A) Schematic depicting experimental design. (B) Structured Illumination super-resolution microscopy (SIM) image of dendritic spines on *+/+, +/RC*, and *+/GS Drd1-Tomato* expressing SPNs, labeled with antibodies to GluA1 (purple), and PSD95 (green). Open arrowheads, GluA1 nanodomains; arrowheads, PSD95 nanodomains. (C) Schematic diagram and object masks depicting GluA1, PSD95, and overlap nanodomains within a dendritic spine. Minimum distance between GluA1-PSD95 is measured from the closest edge of the two nanodomains, as shown. (D, E) Summary graphs and cumulative distribution of the minimum distance between GluA1 and PSD95 nanodomains. (F, G) Summary data and cumulative frequency for the overlap area of GluA1 and PSD95 nanodomains within dendritic spines. Asterisk in D and F reflect statistical significance for Tukey's multiple comparison tests after one-way ANOVA, whereas asterisks in E and G show statistical significance for Bonferroni post-hoc comparisons after Kolmogorov-Smirnov tests. (H) Bar graphs showing the ratio of GluA1-PSD95 overlap area in E relative to PSD95 area, across genotypes. Data are represented as mean ± SEM. (I-K) Correlation plots of overlap in GluA1-PSD95 area versus PSD95 area for *+/+, +/RC*, and *+/GS Drd1-Tomato* expressing SPNs. *p<0.05, **p<0.01, ***p<0.001.

The online version of this article includes the following video, source data, and figure supplement(s) for figure 2:

**Source data 1.** Raw data plotted in graphs in *Figure 2*.

**Figure supplement 1.** LRRK2 mutations alter postsynaptic density area of SPNs.

**Figure supplement 1—source data 1.** Numerical data of the graphs in *Figure 2—figure supplement 1*.

**Figure 2—video 1.** 3D Reconstruction of dendritic spine of an RC dSPN.

https://elifesciences.org/articles/58997#fig2video1

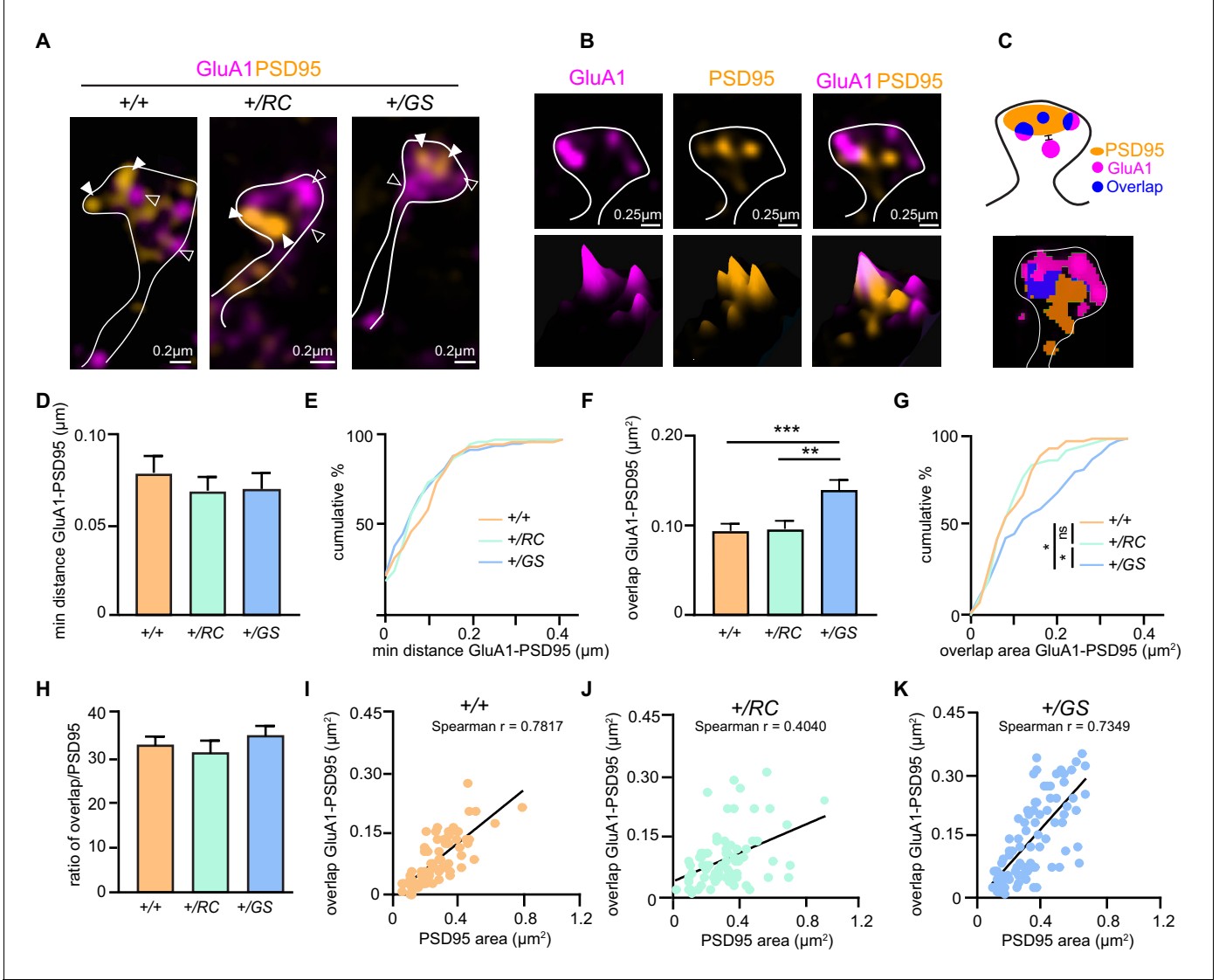

**Figure 3.** LRRK2 mutations alter the nanoscale synaptic organization of iSPNs. (A) SIM image of *+/+*, *+/RC*, and *+/GS* Drd2-eGFP expressing SPNs immunostained with GluA1 (purple), and PSD95 (orange). Open arrowheads, GluA1 nanodomains; arrowheads, PSD95 nanodomains. GFP antibody was used to amplify the Drd2-eGFP signal. (B) Surface intensity through a dendritic spine head in an RC iSPN. (C) Schematic diagram and object masks depicting GluA1, PSD95, overlap nanodomains, and minimum distance between nanodomains within a dendritic spine head. (D) Bar graphs showing the minimum distance between GluA1 and PSD95 nanodomains. (E) Cumulative distribution of data shown in D. (F, G) Summary data and cumulative distribution of the overlap area between GluA1 and PSD95 nanodomains in dendritic spine heads across genotypes. Asterisks in F show statistical significance for Tukey's multiple comparison tests after one-way ANOVA; asterisks in G reflect statistical significance for Bonferroni post-hoc comparisons after Kolmogorov-Smirnov tests. (H) The ratio of overlap area between GluA1 and PSD95 to the PSD95 area for *+/+*, *+/RC*, and *+/GS* iSPNs. (I-K) Correlation plots of overlap areas versus PSD95 area for iSPNs across genotypes. *p<0.05, **p<0.01, ***p<0.001.

The online version of this article includes the following source data for figure 3:

**Source data 1.** Numerica data represented as graphs in *Figure 3*.

We aimed to further investigate the relationship between overlapping nanodomains size and PSD95 area in the iSPNs dendritic spines. In general, PSD95 area correlated with the overlap GluA1-PSD95 domain size in all genotypes (control, Spearman r = 0.7817, p<0.0001, +/RC, r = 0.4040, p=0.0003 and +/GS, r = 0.7349, p<0.0001) (*Figure 3I–K*). Based on *Figure 2I*, we found that for larger PSD95 domains, mutant LRRK2 dSPNs show a tendency towards larger overlap between GluA1 and PSD95. However, this was not the case for iSPNs of *+/RC* genotype. In fact, in *+/RC* iSPNs, some high overlap GluA1-PSD95 domains were characterized by smaller PSD95 areas. This

detailed high-resolution imaging analysis points to different GluA1 subsynaptic distribution in the iSPNs of *+/RC* SPNs, compared to *+/+* and *+/GS*. Taken together, these single synapse imaging findings in pathway identified SPNs demonstrate an increased number of GluA1 receptors in the synapses of *+/RC* dSPNs, pointing toward cell type and mutant-specific LRRK2 functions. These observations suggest highly specific and likely regulated nanoscale organization of GluA1 receptors in SPN dendritic spines.

## Pathway-specific functional alterations in the synapses of R1441C and G2019S SPNs

Since the effects of LRRK2 mutations appear to alter excitatory synapse subunits, next we carried out recordings of pharmacologically isolated miniature excitatory post-synaptic currents (mEPSCs) in acute brain slices from mice of the three genotypes. SPN pathway identity was determined based on the presence or absence of *Drd2-eGFP* (*Kozorovitskiy et al., 2015*). Neurons were held at −70 mV in voltage clamp, and pharmacologically isolated mEPSCs were recorded in the presence of tetrodotoxin (TTX). For dSPNs, the mean frequency of mEPSCs, averaged within each neuron, was decreased relative to controls for both LRRK2 mutations (GFP-, RC 60% of control, GS, 47% of control; two-way ANOVA, genotype main effect, p<0.001, post-hoc tests as noted) (*Figure 4A–B*). For iSPNs, lower mean frequency of mEPSCs, averaged within each neuron, was observed in both RC and GS neurons (GFP-, RC 60% of control, GS, 52% of control; two-way ANOVA, genotype main effect, p<0.001, post-hoc tests as noted). In contrast to the mean frequency of mEPSCs, no significant differences in the mean within-cell response amplitude were observed across all six genotype/ cell type combinations on post-hoc comparisons, despite a modest main effect of genotype (GFP-, RC 107% of control, GS, 92% of control; GFP+, RC 108% of control, GS 94% of control; two-way ANOVA, genotype main effect, p=0.041, N = 13–14 neurons/group) (*Figure 4C*). To evaluate the distribution of individual mEPSC features across genotypes, rather than within-neuron averages, we computed cumulative distributions of inter-event intervals (IEIs) and mEPSC amplitudes, along with histograms of response amplitudes (*Figure 4D–E*). Amplitude data were fitted with a gamma distribution, a non-negative asymmetrical distribution, because the mEPSC amplitudes were not normally or lognormally distributed (*Figure 4*, *Figure 4—figure supplement 1A*), failing D'Agostino and Pearson, Shapiro-Wilk, and KS normality tests, p<0.0001, for raw and log-transformed data. Skew and kurtosis across the 6 datasets were 2.19 ± 0.16 and 6.34 ± 0.1, respectively, appropriate the gamma distribution. Akaike information criterion (AIC) values as goodness-of-fit criteria confirmed superior fit of gamma family distribution over Gaussian. To compare the distributions of mEPSC amplitudes, we estimated mEPSC amplitude means across different genotypes for GFP- and GFP+ cell types using generalized linear modeling (GLM) with a gamma family distribution. mEPSC amplitudes in RC dSPNs were shifted toward larger events, and smaller events in GS dSPNs, with no differences in iSPNs (dSPNs, Bonferroni post-hoc comparisons, p<0.0001; iSPNs, p>0.7) (*Figure 4*, *Figure 4—figure supplement 1B*).

## Overall dendritic spine density and morphology is conserved in mutant LRRK2 SPNs

To investigate whether the observed decrease in the mEPSC frequency was associated with global changes in dendritic spine number and morphology, we relied on identifying pathway-specific SPNs by injecting a Cre-dependent adeno-associated virus (AAV) expressing eGFP virus (AAV8/Flex-GFP) in the striatum of neonatal pups. *Drd1-Cre* and *Adora2a-Cre* mice were crossed with *+/RC* and *+/GS* mice, as well as controls (*Figure 5A*, *Figure 5—video 1*). We then analyzed dendritic fragments of d- and iSPNs to determine dendritic spine density, spine head width, and spine length. No differences in any of these parameters were observed for all six genotype/cell type combinations (two-way ANOVA, n = 9–12 SPNs from 3 to 4 mice/genotype) (*Figure 5B–E*). Further dendritic spine classification showed no difference in dendritic spine type in SPNs of either pathway (two-way ANOVA, n = 9–14 SPNs for 3–4 mice/genotype) (*Figure 5F*). Overall, these data suggest that the functional synaptic alterations (*Figure 4*) observed in *+/RC* and *+/GS* mutant SPNs are not reflected in uniform changes in dendritic spine number or morphology.

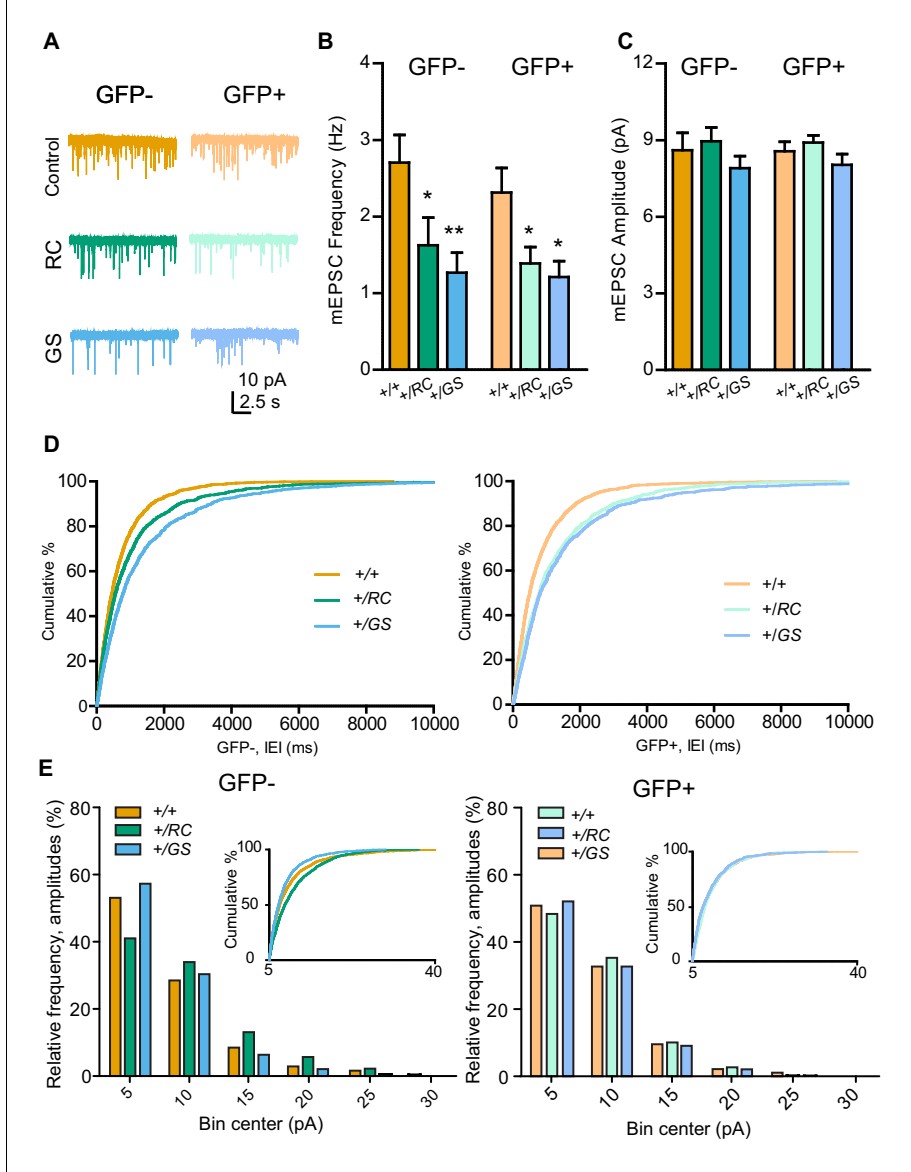

**Figure 4.** Pathway-specific functional alterations of SPN synapses in LRRK2 mutants. (**A**) Example miniature excitatory postsynaptic current (mEPSC) traces from individual neurons of six genotype-pathway combinations. GFP-, dSPNs; GFP+, iSPNs; color, as defined in the figure. Scale bars, 10 pA and 2.5 s. (**B**) Summary graph showing the frequency of pharmacologically isolated mEPSCs, in GFP- and GFP+ SPNs in controls, compared to both RC and GS mutations. Asterisks reflect statistical significance for Bonferroni post hoc comparisons after two-way ANOVA. (**C**) Same as B, but for mEPSC amplitude. (**D**) *Left*, cumulative distribution of inter-event intervals (IEI) for mEPSCs across genotypes for GFP- SPNs. *Right*, same as *left*, but for GFP+ SPNs. (**E**) Binned histograms and cumulative distribution of mEPSC amplitude data. X axis starts at 5 pA, reflecting the amplitude threshold for mEPSC identification. *p<0.05, **p<0.01.

The online version of this article includes the following source data and figure supplement(s) for figure 4:

**Source data 1.** Raw data of the graphs in *Figure 4*.

**Figure supplement 1.** Analyses of mEPSC amplitude distributions.

**Figure supplement 1—source data 1.** Numerical data of *Figure 4—figure supplement 1*.

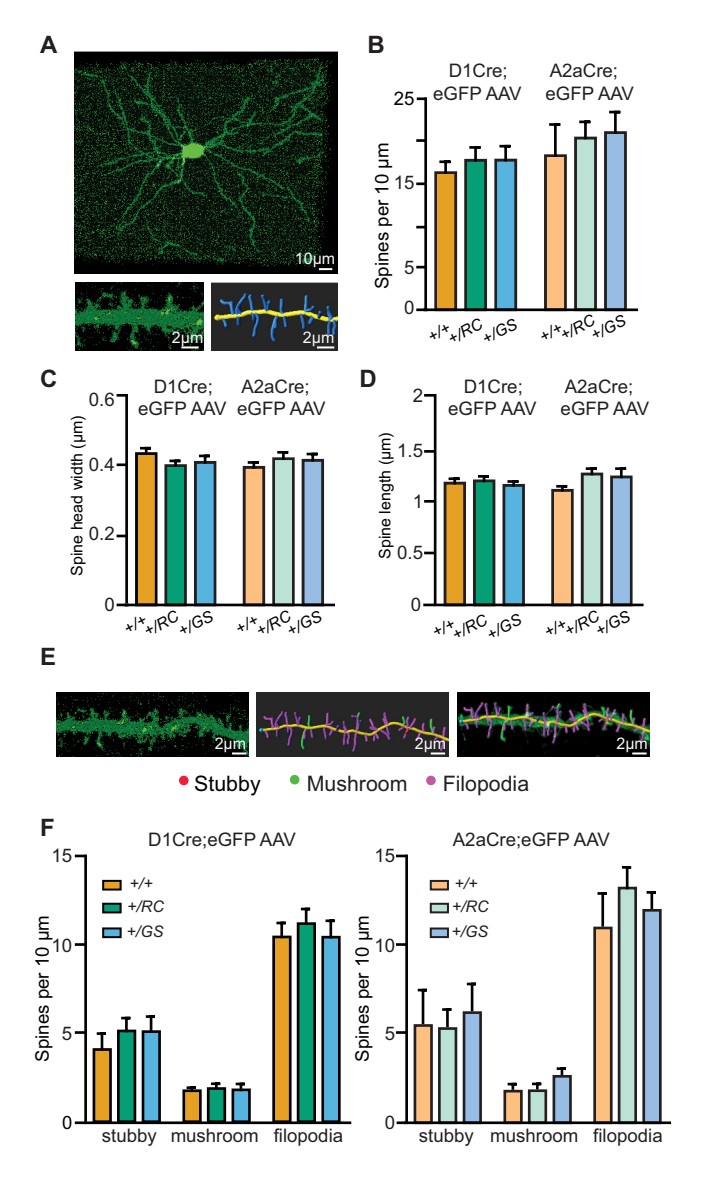

**Figure 5.** Pathway-specific analysis of dendritic spine morphology in LRRK2 mutants. (**A**) Example confocal maximum projection image of a *+/RC;Adora2a-Cre* iSPN expressing AAV8/Flex-GFP. Representative dendritic fragment with dendritic spines and the corresponding 3D Imaris generated filament. (**B**) Summary graph showing the dendritic spine density in pathway-identified SPNs across genotypes. Quantification of dendritic spine head width (**C**), and dendritic spine length (**D**) in d- and iSPNs. (**E**) Confocal maximum projection image and the corresponding 3D Imaris generated filament with classified dendritic spines. Red, stubby; green, mushroom; purple, filopodia. (**F**) *Left*, summary data showing the density of each dendritic spine category in dSPNs. *Right*, same as left, but for iSPNs.

The online version of this article includes the following video and source data for figure 5:

**Source data 1.** Numerical data represented as graphs in *Figure 5*.

**Figure 5—video 1.** 3D Reconstruction of an identified *+/RC;Adora2a-Cre* (Drdr1-, iSPN) neuron transduced with a Cre-dependent AAV eGFP virus (AAV8/Flex-GFP) in neonatal pups.
https://elifesciences.org/articles/58997#fig5video1

## Enhanced single synapse glutamate uncaging-evoked currents in dSPNs of LRRK2 R1441C mutant SPNs

Our results indicate that the abundance of GluA1 subunits is selectively increased in dendritic spines of *+/RC* dSPNs, along with synaptic AMPAR presence and the prevalence of larger amplitude mEPSCs. This is consistent with a possibility that a specific fraction of *+/RC* excitatory synapses are functionally stronger. In order to evaluate the physiology of single dendritic spines in pathway-identified SPNs across three genotypes, we used two-photon dual laser glutamate uncaging and imaging, combined with whole-cell electrophysiology in voltage clamp mode (*Figure 6A*). *Drd2-eGFP* mice were crossed into G2019S and R1441C KI lines, and maintained on a WT background for control groups. SPNs were filled with a cesium-based internal and Alexa 594 for imaging dendrites and dendritic spines at ~910 nm (*Figure 6B–C*); the presence or absence of GFP labeling was used to assign pathway identity. Uncaging evoked EPSCs (uEPSCs) were elicited using 0.5 ms long pulses of 725 nm laser light, directed near dendritic spines (~0.5–1 μm away), in order to drive focal uncaging of MNI-glutamate. Locations of peak responses, sampled for three sites near each dendritic spine, were chosen for data acquisition for every synapse. Recordings were carried out in the presence of blockers of GABAARs, NMDARs, muscarinic receptors, and sodium channels, for isolation of single synapse AMPAR responses. We found that single spine glutamate uEPSCs were increased in amplitude selectively in dSPNs of RC genotype mice (GFP-, *+/RC* 196% of control, GS, 102% of control; one-way ANOVA, $p < 0.05$, post hoc tests, as noted; n = 18–28 dendritic spines/group) (*Figure 6D–F*). For Drd2-eGFP$^+$ SPNs (iSPNs), no uEPSC amplitude differences were observed (GFP+, RC 70% of control, GS, 80% of control; one-way ANOVA, $p = 0.1682$; n = 21–65 dendritic spines/group), suggesting a pathway-specific effect on single synapse function in the RC genotype.

## Discussion

In the present study, we characterized synaptic dysfunctions caused by mutant LRRK2 (RC and GS) in pathway-identified SPNs. To our knowledge, this is the first comparative approach focusing on more than one PD related LRRK2 mutation and employing both global and single synapse approaches for the study of LRRK2 driven striatal remodeling. By studying RC and GS KI mouse lines side by side, we were able to demonstrate that the two most common LRRK2 gain-of-function pathogenic mutations (*Hernandez et al., 2016*) alter the function of SPN excitatory synapses in largely similar ways, with several potentially important differences. The observed changes were frequently stronger for the RC mutations, and more exaggerated in the direct pathway. For example, while whole cell electrophysiology recordings revealed decreases in the mESPCs frequency across both *+/RC* and *+/GS* mice, two-photon dual laser glutamate uncaging and imaging, combined with whole-cell electrophysiology, demonstrated larger amplitude in uEPSCs selectively for *+/RC* dSPNs. This observation paralleled higher synaptic GluA1 receptors levels in the same subtype *+/RC* SPNs (*Figure 6G*).

Based on the current knowledge, it remains unclear why the RC mutation is associated with somewhat more pronounced effects on sculpting striatal synapses. The GS mutation, which has been the main focus of the current literature, is found in the kinase domain of LRRK2, whereas the RC mutation is located in the GTPase domain of the protein (*Paisán-Ruiz et al., 2013*), (*Cookson, 2012*). It has been previously proposed that all pathogenic mutations lead to increased LRRK2 kinase activity, although the mechanism by which RC does this still remains unclear (*Alessi and Sammler, 2018*; *Nguyen and Moore, 2017*). Moreover, it is not known whether aberrant LRRK2 substrate phosphorylation is the predominant pathogenic mechanism in LRRK2 mediated striatal changes. Recent findings have revealed a subset of Rab family of proteins as the long awaited bona fide LRRK2 substrates (*Steger et al., 2016*). Notably, the RC mutation, mainly in the context of heterologous cell lines, leads to higher increase in Rab phosphorylation compared to the GS mutation, which is localized to the kinase domain (*Liu et al., 2018*), (*Purlyte et al., 2018*). Similarly, the evidence of increased synaptic PKA activities was observed in *+/RC* and not *+/GS* striatal synaptic fractions. This increased PKA activity in the RC synapses is expected to positively correlate with elevations in dSPNs signaling (*Zhai et al., 2019*). Specifically, phosphorylation of GluA1 at Ser 845 by PKA facilitates the targeting of the receptors to extrasynaptic membranes and their 'priming' for synaptic insertion and this results in elevated GluA1 synaptic incorporation (*Diering and Huganir, 2018*). Indeed, our biochemical and SIM imaging data in RC mice support this notion. In contrast to dSPNs, PKA-driven signaling requirements for iSPNs are more complex, due to the involvement of

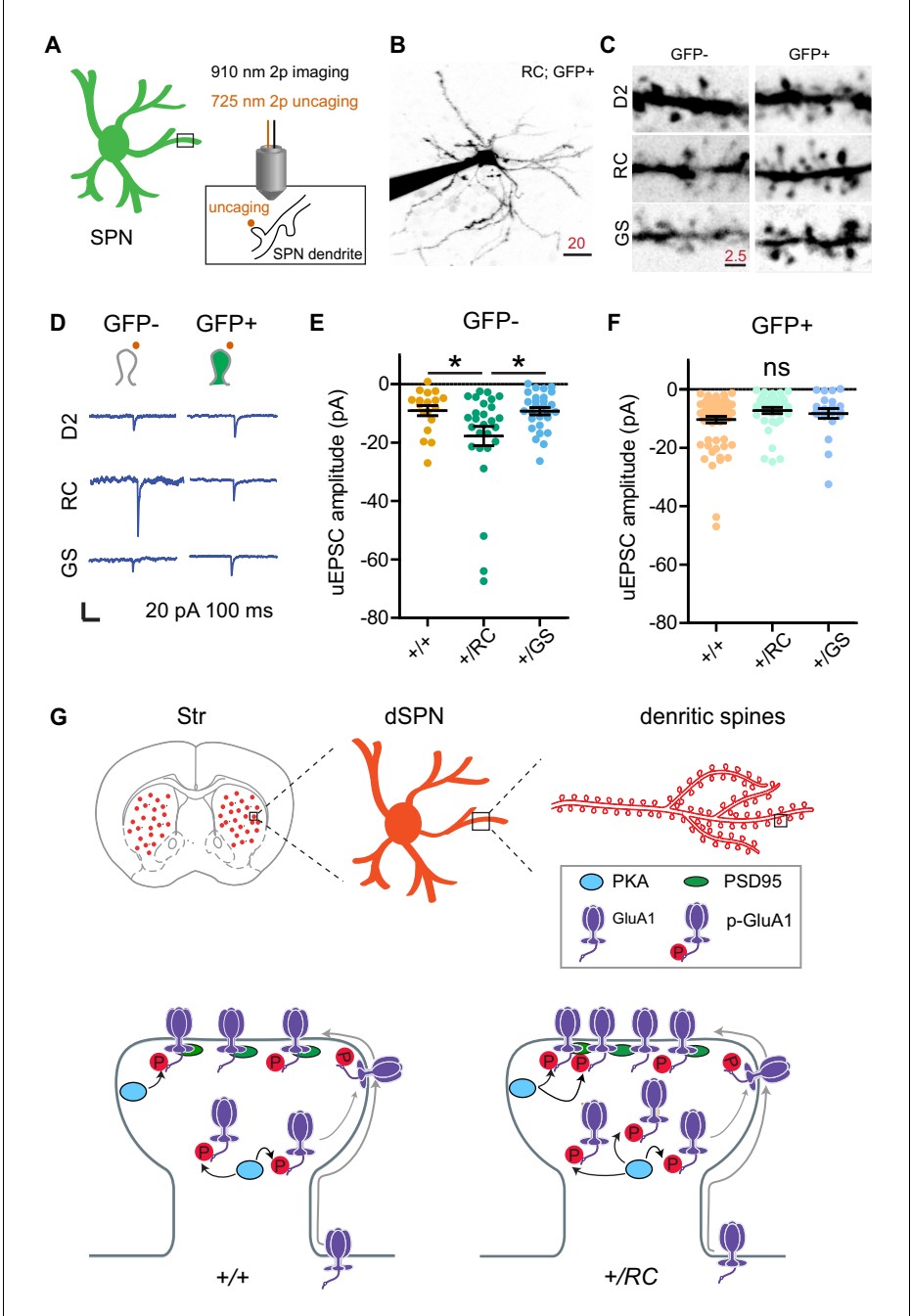

**Figure 6.** LRRK2 RC mutation increases glutamate uncaging-evoked currents in dSPNs. (A) Schematic illustrating experimental design. (B) Example projection of a two-photon laser scanning microscopy stack showing an +/RC; GFP+ SPN (iSPN). Scale bar, 20 μm. (C) Close up images of representative dendrites from SPNs in the six genotype-pathway combinations, shown using inverse greyscale LUT. (D) Example single synapse AMPA-receptor-mediated currents, evoked by focal uncaging of MNI-glutamate. Scale bar, 20 pA and 100 ms. (E) Summary graph showing uncaging-evoked EPSCs (uEPSCs) for GFP-SPNs in controls, RC, and GS mutants. Asterisks reflect statistical significance for Tukey post-hoc comparisons after one-way ANOVA. (F) Same as E, but for GFP+ SPNs. (G) Summary schematic for the study, illustrating changes in synaptic content of glutamate receptors in LRRK2 mutations.

The online version of this article includes the following source data for figure 6:

**Source data 1.** Numerical data of the graphs shown in *Figure 6*.

adenosine. This molecule can be released by neurons or glia (*Surmeier et al., 2007*), (*Zhang et al., 2019*), complicating predictions for PKA involvement in mutant LRRK2 function in iSPNs.

Several prior reports demonstrate a LRRK2 mediated presynaptic regulation of glutamatergic transmission (*Matikainen-Ankney et al., 2016*; *Volta et al., 2017*; *Beccano-Kelly et al., 2015*; *Piccoli et al., 2011*), while emerging evidence emphasizes a postsynaptic role for LRRK2 (*Parisiadou et al., 2014*; *Matikainen-Ankney et al., 2016*). Our previous and current findings showed that LRRK2 contributes to glutamatergic synaptic functions by directing PKA signaling events in SPNs, leading to altered GluA1 synaptic incorporation in the striatum of RC mice (*Parisiadou et al., 2014*). Despite the elevated GluA1 levels in the synapses of *+/RC* dSPNs, there were no alterations in mean mEPSCs amplitude in these neurons. However, cumulative probability analysis demonstrated that the amplitude shifted toward larger values in the *+/RC* dSPNs, suggesting an inhomogeneity of synaptic function, where a fraction of synapses exhibit larger GluA1-dependent responses and single-synapse uncaging-evoked EPSCs. This scenario, on top of overall dampening of mEPSC frequency, is expected to bias mutant dSPN responses to be driven by a narrower set of glutamatergic inputs than is normally the case. Overtime, a small bias could be amplified through the recurrent circuitry of basal ganglia loops (*Kozorovitskiy et al., 2012*). While there are some differences in the age of preparation and in recording conditions, our consistent finding of substantial miniature EPSC frequency decreases for both mutations in pathway-identified SPNs are in disagreement with prior recordings of spontaneous events in the GS mutant mice. One recent report showed elevations in spontaneous EPSCs events frequency in GS KI mice at 1–3 months (*Volta et al., 2017*), while another reported changes in spontaneous but not miniature events with the GS mutation in pre-weaning mutants (*Matikainen-Ankney et al., 2016*), although recordings were primarily performed in a mixed population of direct and indirect pathway SPNs, and neither study thoroughly compared the two mutations. Such differences in spontaneous event frequency point to a potential circuit-level compensatory mechanism that may be driven by disturbances in a subset of SPN excitatory synapses. Surprisingly, in the presence of clear functional and nano-architecture abnormalities, aggregate dendritic spine density and morphology, at least at the ages we evaluated, appeared normal. One possible explanation for this mismatch that requires further investigation is the possibility of higher occurrence of nonfunctional or differently functioning spines in LRRK2 mutant SPNs.

Given the critical role of LRRK2 in excitatory glutamatergic synapses (*Parisiadou et al., 2014*; *Volta et al., 2017*; *Matikainen-Ankney et al., 2016*), a more nuanced understanding of LRRK2-mediated synaptic alterations should facilitate the search for more targeted PD therapies. Here, we focused primarily on interrogating cell type specific LRRK2-based responses. Why does this pathway specificity matter? It is well-established that dopamine loss causes pathway-specific morphological and functional changes in SPNs (*Kravitz et al., 2010*; *Fieblinger et al., 2014*; *Gertler et al., 2008*). Moreover, dSPNs and iSPNs exhibit opposing PKA pathway signaling properties after dopamine receptor activation (*Gerfen and Surmeier, 2011*; *Surmeier et al., 2007*). Given the critical role of LRRK2 in synaptic PKA activities (*Parisiadou et al., 2014*; *Tozzi et al., 2018b*), these mutations are poised to have pathway specific effects and present distinct opportunities for pathway-targeted therapies. Over the past years, a number of transgenic mouse models was generated to study LRRK2 function (*Volta and Melrose, 2017*; *Xiong et al., 2017*); however, the results were found to be inconsistent across studies. Gene-targeted mutant LRRK2 KI mice represent the most physiologically relevant model to investigate LRRK2 mediated alterations and unravel disease mechanisms. Here, we attempt to synthesize and further the existing knowledge by employing a comparative approach focusing on two distinct mutations, combined with a powerful suite of techniques allowing for both global and single synapse study of LRRK2 SPNs.

The clinical phenotypes that define PD arise after a substantial loss of nigrostriatal dopamine signaling. Thus, the identification of pre-symptomatic dysfunctions offers a potential window of opportunity for the development of neuroprotective therapies in PD. Human asymptomatic LRRK2 mutation carriers represent an appropriate population to define these preclinical symptoms. Indeed, emerging evidence suggests that asymptomatic LRRK2 mutation carriers do show subtle motor and non-motor symptoms, including alterations of corticostriatal circuit organization, in comparison to asymptomatic non-carriers (*PPMI Investigators et al., 2020*; *LRRK2 Ashkenazi Jewish Consortium et al., 2015*). Accordingly, cellular and synaptic dysfunctions in etiologically relevant LRRK2 mutant KI mice allow investigations of the early events that precede neuronal death and may be predictive

of future dysfunction. Several studies have suggested a central role for LRRK2 in striatal SPNs (*Volta et al., 2017*), (*Parisiadou et al., 2014*; *Xenias, 2020*). Here, we show that the presence of LRRK2 mutations can influence SPN function in a pathway-specific context, in the absence of dopamine neurodegeneration. Examining PD-relevant cellular functions in the absence of DA depletion represents a move away from the classical ways of studying PD in small rodent models, where basal ganglia cellular and network properties have been examined using several methods for dopamine depletion (*Gerfen and Surmeier, 2011*; *Kreitzer and Malenka, 2008*). While there is evidence that at first iSPNs show the primary structural and functional synaptic and dendritic changes in response to dopamine loss (*Gerfen, 2006*; *Gertler et al., 2008*; *Day et al., 2006*; *Villalba et al., 2009*), with time both SPNs types undergo substantial adaptations (*Suarez et al., 2018*), (*Gagnon et al., 2017*). Here, we find that some early changes in LRRK2 mutant dSPNs – including those observed at the single synapse level – are poised to contribute to the fragility of the system to lower levels of DA loss/variation. Because the well-described recurrent circuitry of the basal ganglia (*Kozorovitskiy et al., 2012*), (*Kravitz et al., 2010*; *Oldenburg and Sabatini, 2015*) enables the output of direct and indirect pathways to feed back to regulate cortico-striatal glutamate release, minor synaptic defects in LRRK2 mutant SPNs could be amplified over time, contributing to the life-time risk of the disorder. The presence of early corticostriatal alterations in LRRK2 carriers (*LRRK2 Ashkenazi Jewish Consortium et al., 2015*; *Vilas et al., 2015*; *the Barcelona LRRK2 Study Group et al., 2016*), along with the evidence for finer scale synaptic dysfunctions reported in this study, open the possibilities for future personalized medicine approaches that take into account the specific LRRK2 mutation and emphasize protecting the health of striatal neurons prior to potential DA losses.

## Materials and methods

### Mouse strains and genotyping

All mouse related experiments followed the guidelines approved by the Northwestern University Animal Care and Use Committee. Young adult male and female mice (postnatal days 30–50) were used in this study. Approximately equal numbers of males and females were used for every experiment. All mice were group-housed, with standard feeding, light-dark cycle, and enrichment procedures. C57BL/6 (wild-type), *RC* (*Tong et al., 2009*)*, and GS* (*Yue et al., 2015*) heterozygous knock in mice were used for subcellular fractionation experiments. For electrophysiological and imaging approaches *RC* or *GS* mice were crossed with *Drd1-dTomato* and/or *Drd2-eGFP* mice. Heterozygotes *for RC and GS* allele and hemizygotes for *Drd1-dTomato* and/or *Drd2-eGFP* were used in all experiments. For dendritic spine analysis, hemizygous *Drd1-Cre*, and *Adora2a-Cre* mice crossed with RC and GS KI mice were used. All animals were backcrossed to C57BL/6 for several generations.

### Subcellular fractionation and western blot analysis

Subcellular fractionation of mouse striatum was performed as previously described (*Bermejo et al., 2014*), (*Parisiadou et al., 2014*; *Figure 1B*). Specifically, mouse striata were dissected (three striata per experiment were pooled) and rapidly homogenized in four volumes of ice-cold Buffer A (0.32 M sucrose, 5 mM HEPES, pH7.4, 1 mM $MgCl_2$, 0.5 mM $CaCl_2$) supplemented with Halt protease and phosphatase inhibitor cocktail (Thermo) using a Teflon homogenizer (12 strokes). Homogenized brain extract was centrifuged at 1400 *g* for 10 min. Supernatant (S1) was saved and pellet (P1) was homogenized in buffer A with a Teflon homogenizer (five strokes). After centrifugation at 700 *g* for 10 min, the supernatant (S1′) was pooled with S1. Pooled S1 and S1′ were centrifuged at 13,800 *g* for 10 min to the crude synaptosomal pellet (P2) and the supernatant (S2). P2 was resuspended in Buffer B (0.32 M sucrose, 6 mM Tris, pH 8.0) supplemented with protease and phosphatase inhibitors cocktail with Teflon homogenizer (five strokes) and was carefully loaded onto a discontinuous sucrose gradient (0.8 M/1 M/1.2 M sucrose solution in 6 mM Tris, pH 8.0) with a Pasteur pippete, followed by centrifugation in a swinging bucket rotor for 2 hr at 82,500 *g*. The synaptic plasma membrane fraction (SPM) in the interphase between 1 M and 1.2 M sucrose fractions was collected using a syringe and transferred to clean ultracentrifuge tubes. 6 mM Tris buffer was added to each sample to adjust the sucrose concentration from 1.2 M to 0.32 M and the samples were centrifuged in a swinging bucket rotor at 200,000 g for 30 min. The supernatant was removed and discarded and the

SPM pellet was resuspended in 500 µl of 6 mM Tris/2 mM EDTA/0.5% Triton X-100 solution and rotated for 30 min at 4°C. In turn, the samples were centrifuged at 32,800 $g$ for 20 min. The supernatant contains the Triton-soluble fraction (TSF), whereas the pellet represents the postsynaptic pellet (PSD).

S1, P2, presynaptic and PSD fractions were separated by 4–12% NuPage Bis-Tris PAGE (Invitrogen) and transferred to membranes using the iBlot nitrocellulose membrane Blotting system (Invitrogen) by following manufacture protocol. Primary antibodies specific for GluA1 (Cell Signaling Technology #13185), pGluA1 (Cell Signaling Technology #8084), phospho-PKA substrates (Cell Signaling Technologies #9624), phospho-Rab8A (Abcam, ab188574), total Rab 8A (Abcam, ab230260), phospho-Rab10 (Abcam ab230261) and total Rab10 (Cell Signaling Technologies #8127) as well as secondary anti-mouse and anti-rabbit (Thermo Fischer Scientific) Membranes were incubated with Immobilon ECL Ultra Western HRP Substrate (Millipore) for 3 min prior to image acquisition. Chemiluminescent blots were imaged with iBright CL1000 imaging system (Thermo Fisher Scientific). For quantitative analysis, images were analyzed using iBright Analysis Software (Thermo Fisher Scientific).

## Primary neuronal cultures

Primary corticostriatal co-cultures were prepared as described previously (*Parisiadou et al., 2014*), (*Tian et al., 2010*). In brief, striatal cultures were prepared from P0 pups of *Drd1-Tomato* or *Drd2-eGFP* mice crossed with mutant KI LRRK2 lines (*G2019S* and *R1441C*). Tissues were digested by papain (Worthington Biochemical Corporation) and the striatal and cortical cells were mixed at a ratio of 1:2. The neurons were placed on coverslips with plating medium (Medium I) containing Basal Eagle Medium (Sigma-Aldrich) supplemented with 1 x GlutaMAX (Gibco), 1 × B27 (Gibco), 1 x N-2 (Gibco), 1 x Antibiotic Antimycotic (Sigma-Aldrich), 5% horse serum (Gibco) and 5% FBS (Gibco) at a density of $4 \times 10^5$ for about one hour. After initial plating, medium was changed to Medium I without the horse serum and the FBS supplemented with 2.5 µM arabinosylcytosine (Sigma-Aldrich) (Medium II). Half of the medium was changed with fresh Medium II every 7 days; experiments were conducted 28–30 days after plating.

## SIM imaging and analysis

Multichannel SIM images were obtained with a Nikon Structured Illumination super-resolution microscope using a 100x, 1.4 NA objective as previously described (*Smith et al., 2014*). The acquisition was set to 10 MHz, 16 bit depth with EM gain and no binning. Exposure was between 100 and 500 ms and the EM gain multiplier restrained below 1500. Conversion gain was held at 1x. Laser power was adjusted to keep LUTs within the first quarter of the scale. Single images were processed and analyzed using Nikon Elements software. Reconstruction parameters (Illumination Modulation Contrast, High Resolution Noise Suppression, and Out of Focus Blur Suppression) (0.94, 0.98, and 0.07) for GluA1/PSD95 and (1.2, 3.0, and 0.07) for GFP/tdTomato were kept consistent across all acquisitions and experiments. Single spine analyses were carried out on 72–95 spines across 9–12 neurons per genotype. Images of dendritic fragments were collected from secondary to tertiary dendrites across genotypes. All the spines on the fragments were analyzed. GFP/tdTomato signal (Thermo Fischer Scientific, #A10262/#M11217) was used to generate a mask to identify the spine shape, subsequent analysis were done with the nanodomains of GluA1 (NeuroMab #75327) and PSD95 (Thermo Fischer Scientific, #51–6900) within the mask. 3D reconstructions in a single plane used nine images captured with 2D SIM and reconstructed with 3D SIM utilizes 15 to increase xy and z resolution were generated by Nikon Element and the illumination modulation contrast was set automatically by the software. Nikon Elements software (general analysis) was used for quantification of colocalization of GluA1 and PSD95 proteins. The images were thresholded for each channel and kept constant across experiments. The regions of interest (puncta) were outlined, and total immunofluorescence number and binary area for each region per channel were measured automatically. Regions in one channel were overlayed on the other channel. The minimum distances between puncta (e.g. PSD95 to GluA1) were measured from surface to surface automatically by the software.

## Intracranial injections

Intracranial injections were performed as previously described (*Kozorovitskiy et al., 2012*). Briefly, P4-5-day-old pups were placed into a stereotaxic frame under cryoanesthesia. 200 nl of AAV8/Flex-GFP virus (6.2*10$^{12}$ used at 1:3 dilution, UNC Vector Core, Chapel Hill, NC) were delivered into the dorsal striatum at a rate of 100 nl min$^{-1}$ using microprocessor-based controller, Micro4 (WPI). In order to ensure targeting of the dorsal striatum the needle was placed 1 mm anterior to midpoint between ear and eye, 1.5 mm from midline and 1.8 mm ventral to brain surface.

## Confocal microscopy and dendritic spine analysis

Confocal images of fixed 80-µm-thick brain sections of P30 pups injected with the AAV8/Flex-GFP virus were obtained with the Nikon A1R scope. Fluorescence projection images of dendrites and the corresponding spines were acquired with a 60x oil immersion objective (NA = 1.4) at 0.1 µm intervals with 1024 × 1024 pixel resolution. For each genotype, 2–4 segments per neuron, 9–14 neurons from 3 to 4 animals were used to generate z-stacks. Fragments between secondary to tertiary dendrites without overlap with other neurons or discontinuous were chosen for analysis. Dendritic spine density and morphology was performed using Imaris 9.21 software (Bitplane, Concord, USA). Images of dendritic fragments were collected from secondary to tertiary dendrites, for tracing of the dendritic fragments the autopath mode of the filament tracer and default settings were selected. The following settings were used for spine detection: 0.5 µm minimal head size, 1.8 µm maximum length, seed point threshold approx. 10, no branched spines were allowed. Spine detection was manually corrected if necessary. Classification of spines into stubby, mushroom-like and filopodia was performed using the Imaris XTension classify spines with following definitions: stubby: spine length <0.75 µm; mushroom: spine length <3.5 µm, spine head width >0.35 µm and spine head width >spine neck width; filopodia: when it did not fit the criteria mentioned above (*Schier et al., 2017*).

## Electrophysiological recordings

Two hundred and fifty µm thick acute coronal brain slices were prepared as described previously (*Kozorovitskiy et al., 2015*), (*Xiao et al., 2017*) and incubated in artificial cerebral spinal fluid (ACSF) containing (in mM) 127 NaCl, 2.5 KCl, 25 NaHCO$_3$, 1.25 NaH$_2$PO$_4$, 2.0 CaCl$_2$, 1.0 MgCl$_2$, and 25 glucose (osmolarity ~310 mOsm/L). Slices were recovered at 34°C for 15 min, followed by 30 min at RT, and transferred to a recording chamber perfused with oxygenated ACSF at a flow rate of 2–3 mL/min. Whole-cell recordings were obtained from dorsolateral striatal neurons visualized under infrared Dodt or DIC contrast video microscopy using patch pipettes of ~4–6 MΩ resistance. Drd2 BAC GFP signal visualized under epifluorescence was used to target iSPNs, while Drd2 GFP-negative neurons with spiny dendrites and electrophysiological properties of SPNs (holding current, input resistance) were considered dSPNs (*Gagnon et al., 2017*), (*Gittis et al., 2010*). Both fluorescent and non-fluorescent neurons were targeted for recording. Internal solution consisted of (in mM): 120 CsMeSO$_4$, 15 CsCl, 10 HEPES, 2 QX-314 Chloride, 2 Mg-ATP, 0.3 Na-GTP, 1 EGTA (pH ~7.2,~295 mOsm). Morphology was confirmed using 20 µM Alexa 594 in the recording pipette. Recordings were made using a Multiclamp 700B amplifier (Molecular Devices). Data were sampled at 10 kHz and filtered at 4 kHz, acquired in MATLAB (MathWorks). Series resistance, measured with a 5 mV hyperpolarizing pulse in voltage clamp, averaged under 20 MΩ and was left uncompensated. Miniature EPSCs were recorded from voltage clamped SPNs held at −70 mV in the absence of stimulation. Over 2 min of recording per neuron was used for analyses. For recordings of miniature EPSCs, 50 µM gabazine, 10 µM scopolamine, 10 µM CPP, and 1 µM TTX were added to the ACSF.

## Two-photon imaging and uncaging

Two-photon laser-scanning microscopy and two-photon laser photoactivation were accomplished on a modified Scientifica microscope with a 60X (1.0 NA) objective. Two mode-locked Ti:Sapphire lasers (Mai-Tai eHP Deep See and Mai-Tai eHP; Spectra Physics) were separately tuned, with beam power controlled by independent Pockels cells (ConOptics). The beams were separately shuttered, recombined using a polarization-sensitive beam-splitting cube (Thorlabs), and guided into the same galvanometer scanhead (Cambridge). The Mai Tai eHP Deep See was tuned to ~910 nm for excitation of genetically encoded GFP and Alexa 594, and the Mai Tai eHP was tuned to 725 nm for photoactivation of recirculated caged MNI-L-glutamate (2 mM, Tocris) (*Xiao et al., 2018*; *Banala et al., 2018*).

Neurons in the dorsolateral striatum were targeted for recording, as for other electrophysiology assays, with secondary and tertiary dendrites targeted for uncaging. All 2P uncaging voltage clamp recordings were made at a holding potential of −70 mV. Internal solution contained (in mM) 115 K-gluconate, 20 KCl, 4 MgCl$_2$, 10 HEPES, 4 Mg-ATP, 0.3 Na-GTP, 7 phosphocreatine, 0.1 EGTA, (pH 7.2, 290 mOSm). Alexa Fluor 594 (10–20 µM) was added to the internal solution to visualize cell morphology for uncaging/imaging with physiology experiments. Uncaging evoked EPSCs were elicited by 0.5 ms pulses of 725 nm laser light (~20 mW at the focal plane). Up to three locations in a single field of view were stimulated (1 s apart) in a single sweep. Stimulation of a single location occurred with a minimum 10 s ISI. A spot diameter of ≤0.8 µm, based on measurements of 0.5 µm beads (17152–10; Polysciences Inc) was used for all two-photon laser flash photolysis experiments. Two GaAsP photosensors (Hamamatsu, H7422) with 520/28 nm band pass filters (Semrock), mounted above and below the sample, were used for imaging fluorescence signals. A modified version of ScanImage was used for data acquisition (*Pologruto et al., 2003*), (*Kozorovitskiy et al., 2015*).

## Statistical analyses

Group statistical analyses were done using GraphPad Prism seven software (GraphPad, LaJolla, CA). For n sizes, both the number of trials recorded and the number of animals are provided. All data are expressed as mean + SEM, or individual plots. For two-group comparisons, statistical significance was determined by two-tailed Student's t-tests. For multiple group comparisons, one-way analysis of variance (ANOVA) tests were used for normally distributed data, followed by post-hoc analyses. For non-normally distributed data, non-parametric tests for the appropriate group numbers were used, such as the Mann-Whitney test. For the analysis of mEPSC amplitude distributions, data were fit with a gamma distribution, a non-negative asymmetrical distribution, because the mEPSC amplitude data were not normally or lognormally distributed but displayed appropriate skew and kurtosis to match the gamma distribution. Generalized linear modeling (GLM) with a gamma family distribution was used to estimate and compare mEPSC amplitudes, with Bonferroni corrections for multiple posthoc comparisons. All mEPSC distribution analyses were done in R, using fitdistrplus library and the emmeans package. Spearman correlation was used to detect correlation between two groups of data. $p < 0.05$ was considered statistically significant.

## Acknowledgements

This work was supported by Michael J Fox Foundation for Parkinson's Research (LP), NIH R01 NS097901 (LP), NINDS R01 NS107539 (YK), Rita Allen Foundation Scholar Award (YK), Searle Scholar Award (YK), and Beckman Young Investigator Award (YK). NB was supported by NINDS F32 NS103243 and VD by a predoctoral award from the American Heart Association (19PRE34380056). SIM imaging was performed at the Center for Advanced Microcopy, Northwestern University, supported by the NIH 1S10OD016342-01 and NCI CCSG P30 CA060553. The authors are grateful to Lindsey Butler for mouse colony management. We also thank Dr. Heather Melrose for providing the LRRK2 G2019S knock-in mice.

## Additional information

### Funding

| Funder | Grant reference number | Author |
| --- | --- | --- |
| National Institute of Neurological Disorders and Stroke | R01NS097901 | Loukia Parisiadou |
| Michael J. Fox Foundation for Parkinson's Research | LRRK2 Challenge | Loukia Parisiadou |
| National Institute of Neurological Disorders and Stroke | R01NS107539 | Yevgenia Kozorovitskiy |
| Rita Allen Foundation | Rita Allen Scholar Award | Yevgenia Kozorovitskiy |
| Kinship Foundation | Searle Scholar Award | Yevgenia Kozorovitskiy |
| Arnold and Mabel Beckman | Beckman Young | Yevgenia Kozorovitskiy |

| Foundation | Investigator Award | |
|---|---|---|
| National Institute of Neurological Disorders and Stroke | F32NS103243 | Nicholas Bannon |
| American Heart Association | 19PRE3438005 | Vasin Dumrongprechachan |

The funders had no role in study design, data collection and interpretation, or the decision to submit the work for publication.

## Author contributions

Chuyu Chen, Conceptualization, Data curation, Formal analysis, Validation, Investigation, Visualization, Methodology, Writing - original draft, Writing - review and editing; Giulia Soto, Formal analysis, Validation, Investigation, Visualization, Methodology, Writing - review and editing; Vasin Dumrongprechachan, Formal analysis, Methodology, Writing - review and editing; Nicholas Bannon, Data curation, Formal analysis, Validation, Visualization, Methodology, Writing - review and editing; Shuo Kang, Formal analysis, Methodology, Writing - original draft, Writing - review and editing; Yevgenia Kozorovitskiy, Loukia Parisiadou, Conceptualization, Resources, Data curation, Software, Formal analysis, Supervision, Funding acquisition, Validation, Investigation, Visualization, Methodology, Writing - original draft, Project administration, Writing - review and editing

## Author ORCIDs

Chuyu Chen https://orcid.org/0000-0001-5666-5173
Vasin Dumrongprechachan https://orcid.org/0000-0001-5890-6778
Yevgenia Kozorovitskiy https://orcid.org/0000-0002-3710-1484
Loukia Parisiadou https://orcid.org/0000-0002-2569-4200

## Ethics

Animal experimentation: All mouse experiments were approved by Northwestern University Animal Care and Use Committee (Approved protocol numbers IS000035451, IS00000838, and 00009022).

## Decision letter and Author response

Decision letter https://doi.org/10.7554/eLife.58997.sa1
Author response https://doi.org/10.7554/eLife.58997.sa2

# Additional files

## Supplementary files

• Transparent reporting form

## Data availability

All data generated during this study are included in the manuscript and supporting files.

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

# Appendix 1

**Appendix 1—key resources table**

| Reagent type (species) or resource | Designation | Source or reference | Identifiers | Additional information |
|---|---|---|---|---|
| Antibody | Phospho-PKA Substrate (RRXS*/T*) (100G7E) rabbit monoclonal | Cell Signaling Technology | RRID:AB_331817 | WB (1:1000) |
| Antibody | AMPA Receptor 1 (GluA1) (D4N9V) rabbit monoclonal | Cell Signaling Technology | RRID:AB_2732897 | WB (1:1000) |
| Antibody | Phospho-AMPA Receptor 1 (GluA1) (Ser845) (D10G5) rabbit monoclonal | Cell Signaling Technology | RRID:AB_10860773 | WB (1:1000) |
| Antibody | Synaptophysin rabbit polyclonal | Cell Signaling Technology | RRID:AB_1904154 | WB (1:1000) |
| Antibody | Anti-RAB8A antibody [EPR14873] rabbit monoclonal | Abcam | RRID:AB_2814989 | WB (1:1000) |
| Antibody | RAB8A (phospho T72) [MJF-R20] rabbit monoclonal | Abcam | RRID:AB_2814988 | WB (1:1000) |
| Antibody | Rab10 (D36C4) XP Rabbit mAb rabbit monoclonal | Cell signaling technology | RRID:AB_10828219 | WB (1:1000) |
| Antibody | Anti-RAB10 (phospho T73) antibody [MJF-R21] rabbit monoclonal | Abcam | RRID:AB_2811274 | WB (1:1000) |
| Antibody | HOMER1 polyclonal antibody rabbit polyclonal | Proteintech | RRID:AB_2295573 | WB (1:1000) |
| Antibody | GluA1/GluR1 glutamate receptor clone N355/1 mouse monoclonal | UC Davis/NIH NeuroMab Facility | RRID:AB_2315840 | IF (1:300) |
| Antibody | PSD-95 monoclonal (6G6-1C9) mouse monoclonal | Invitrogen | RRID:AB_325399 | WB (1:1000) IF (1:300) |
| Antibody | PSD-95 polyclonal rabbit polyclonal | Invitrogen | RRID:AB_87705 | IF (1:300) |
| Antibody | GFP chicken polyclonal | Invitrogen | RRID:AB_2534023 | IF (1:1000) |
| Antibody | mCherry (16D7) rat monoclonal | Invitrogen | RRID:AB_2536611 | IF (1:1000) |
| Antibody | Goat anti-Mouse IgG (H+L) Secondary Antibody, HRP | Invitrogen | RRID:AB_2533947 | WB (1:5000) |
| Antibody | Goat anti-Rabbit IgG (H+L) Secondary Antibody, HRP | Invitrogen | RRID:AB_2533967 | WB (1:5000) |
| Antibody | Donkey anti-Mouse IgG (H+L) Highly Cross-Adsorbed Secondary Antibody, Alexa Fluor 647 | Invitrogen | RRID:AB_162542 | IF (1:300) |
| Antibody | Donkey anti-Rabbit IgG (H+L) Highly Cross-Adsorbed Secondary Antibody, Alexa Fluor 568 | Invitrogen | RRID:AB_2534017 | IF (1:300) |
| Antibody | Goat anti-Rat IgG (H+L) Cross-Adsorbed Secondary Antibody, Alexa Fluor 568 | Invitrogen | RRID:AB_2534121 | IF (1:300) |

*Continued on next page*

*Appendix 1—key resources table continued*

| Reagent type (species) or resource | Designation | Source or reference | Identifiers | Additional information |
|---|---|---|---|---|
| Antibody | Goat anti-Chicken IgY (H+L) Secondary Antibody, Alexa Fluor 488 | Invitrogen | RRID:AB_2534096 | IF (1:300) |
| Antibody | Donkey anti-Rabbit IgG (H+L) Highly Cross-Adsorbed Secondary Antibody, Alexa Fluor 488 | Invitrogen | RRID:AB_2535792 | IF (1:300) |
| Other | Fetal Bovine Serum | Sigma-Aldrich | F0926 | 5% |
| Other | Horse Serum, heat inactivated, New Zealand origin | Gibco | 26050088 | 5% |
| Peptide, recombinant protein | Laminin Mouse Protein, Natural | Gibco | 23017015 | 10 µg/ml |
| Chemical compound, drug | Poly-D-lysine hydrobromide | Sigma | P0899 | 50 µg/ml |
| Chemical compound, drug | GlutaMAX Supplement (100x) | Gibco | 35050061 | |
| Chemical compound, drug | B-27 Supplement (50X), serum free | Gibco | 17504044 | |
| Chemical compound, drug | N-2 Supplement (100X) | Gibco | 17502001 | |
| Chemical compound, drug | Antibiotic Antimycotic Solution (100×), Stabilized | Sigma-Aldrich | A5955 | |
| Chemical compound, drug | Basal Medium Eagle | Sigma-Aldrich | B1522 | |
| Chemical compound, drug | Cytosine β-D-arabinofuranoside | Sigma-Aldrich | C1768 | 2.5 µM |
| Chemical compound, drug | PDS Kit, Papain Vial | Worthington Biochemical Corporation | LK003178 | |
| Chemical compound, drug | DNase Vial (D2) | Worthington Biochemical Corporation | LK003172 | |
| Chemical compound, drug | Sucrose | Sigma-Aldrich | S7903 | |
| Chemical compound, drug | HEPES solution | Sigma-Aldrich | H0887 | |
| Chemical compound, drug | UltraPure 1M Tris-HCl, pH 8.0 | Invitrogen | 15568025 | |
| Chemical compound, drug | MgCl2 | Sigma-Aldrich | M8266 | |
| Chemical compound, drug | Calcium chloride solution | Sigma-Aldrich | 21115 | |
| Chemical compound, drug | Triton X-100 | Sigma-Aldrich | X100 | |
| Chemical compound, drug | Ethylenediaminetetraacetic acid disodium salt solution | Sigma-Aldrich | E7889 | |
| Chemical compound, drug | Halt Protease and Phosphatase Inhibitor Cocktail, EDTA-free (100X) | Thermo Scientific | 78441 | |

*Continued on next page*

*Appendix 1—key resources table continued*

| Reagent type (species) or resource | Designation | Source or reference | Identifiers | Additional information |
|---|---|---|---|---|
| Chemical compound, drug | Tris Buffered Saline (10x) | Sigma-Aldrich | T5912 | |
| Chemical compound, drug | TWEEN 20 | Sigma-Aldrich | P1379 | |
| Chemical compound, drug | MNI-L-glutamate | Tocris | 1490 | |
| Chemical compound, drug | Alexa Fluor 594 | Thermo Fisher Scientific | A10438 | |
| Chemical compound, drug | Sodium chloride | Sigma-Aldrich | S3014 | |
| Chemical compound, drug | Potassium chloride | Sigma-Aldrich | P9541 | |
| Chemical compound, drug | Sodium bicarbonate | Sigma-Aldrich | S5761 | |
| Chemical compound, drug | Sodium phosphate monobasic | Sigma-Aldrich | S3139 | |
| Chemical compound, drug | Calcium chloride | Sigma-Aldrich | C5670 | |
| Chemical compound, drug | D-(+)-Glucose | Sigma-Aldrich | G7021 | |
| Chemical compound, drug | Cesium methanesulfonate | Sigma-Aldrich | C1426 | |
| Chemical compound, drug | Cesium chloride | Sigma-Aldrich | C3032 | |
| Chemical compound, drug | HEPES | Sigma-Aldrich | 54457 | |
| Chemical compound, drug | Ethylene glycol-bis (2-aminoethylether)-N,N,N',N'-tetraacetic acid | Sigma-Aldrich | E3889 | |
| Chemical compound, drug | Gabazine/SR 95531 hydrobromide | Tocris | 1262 | |
| Chemical compound, drug | Scopolamine hydrobromide | Tocris | 1414 | |
| Chemical compound, drug | Phosphocreatine disodium salt hydrate | Sigma-Aldrich | P7936 | |
| Chemical compound, drug | QX-314 Chloride | Tocris | 2313 | |
| Chemical compound, drug | Adenosine 5'-triphosphate magnesium salt | Sigma-Aldrich | A9187 | |
| Chemical compound, drug | Guanosine 5'-triphosphate sodium salt hydrate | Sigma-Aldrich | 51120 | |
| Chemical compound, drug | (R)-CPP | Tocris | 0247 | |
| Chemical compound, drug | Tetrodotoxin | Tocris | 1078 | |
| Chemical compound, drug | Potassium gluconate | Sigma-Aldrich | 1550001 | |
| Commercial assay or kit | iBlot Transfer Stack, nitrocellulose, regular size | Invitrogen | IB301031 | |

*Continued on next page*

*Appendix 1—key resources table continued*

| Reagent type (species) or resource | Designation | Source or reference | Identifiers | Additional information |
|---|---|---|---|---|
| Commercial assay or kit | NuPAGE 4–12% Bis-Tris Protein Gels, 1.5 mm, 15-well | Invitrogen | NP0336BOX | |
| Commercial assay or kit | NuPAGE MES SDS Running Buffer (20X) | Invitrogen | NP0002 | |
| Commercial assay or kit | NuPAGE Antioxidant | Invitrogen | NP0005 | |
| Commercial assay or kit | NuPAGE Transfer Buffer (20X) | Invitrogen | NP0006 | |
| Commercial assay or kit | NuPAGE LDS Sample Buffer (4X) | Invitrogen | NP0008 | |
| Commercial assay or kit | NuPAGE Sample Reducing Agent (10X) | Invitrogen | NP0009 | |
| Commercial assay or kit | Pierce BCA Protein Assay Kit | Thermo Scientific | 23225 | |
| Commercial assay or kit | Restore Western Blot Stripping Buffer | Thermo Scientific | 21059 | |
| Other | ProLong Diamond Antifade Mountant with DAPI | Invitrogen | P36971 | |
| Other | BLUeye Prestained Protein Ladder | Sigma-Aldrich | 94964 | |
| Commercial assay or kit | Immobilon ECL Ultra Western HRP Substrate | Millipore | WBULS0500 | |
| Software, algorithm | GraphPad Prism 7 | GraphPad Software Inc | RRID:SCR_002798 | |
| Software, algorithm | NIS-elements 5.10 | Nikon Instruments Inc | RRID:SCR_014329 | |
| Software, algorithm | Imaris 9.21 | Bitplane Inc | RRID:SCR_007370 | |
| Software, algorithm | iBright Analysis Software | Thermo Scientific | RRID:SCR_017632 | |
| Software, algorithm | MATLAB | MathWorks | RRID:SCR_001622 | |
| Software, algorithm | FIJI | *Schindelin et al., 2012* | http://fiji.sc/; RRID:SCR_002285 | |
| Strain, strain background | rAAV8/Flex-GFP | UNC GTC vector core | Lot, AV4910B | |

