## [Decision Letter]

**Acceptance summary:**

The reviewers felt that your study nicely integrated several approaches to compare indirect and direct striatal output pathways, and to show distinctions in these pathways in 2 LRRK2 mutations.

**Decision letter after peer review:**

Thank you for submitting your article "Pathway-specific deregulation of striatal excitatory synapses in LRRK2 mutations" for consideration by *eLife*. Your article has been reviewed by two peer reviewers, including Carl Lupica as the Reviewing Editor and Reviewer #1, and the evaluation has been overseen by John Huguenard as the Senior Editor. The following individual involved in review of your submission has agreed to reveal their identity: Michael J Beckstead (Reviewer #2).

The reviewers have discussed the reviews with one another and the Reviewing Editor has drafted this decision to help you prepare a revised submission.

Summary:

This is a report on the effects of 2 common LRRK2 gain of function mutations expressed in mouse lines on the properties of striatal neurons comprising either indirect (iSPNs) or direct pathway projections (dSPNs). The authors use structured illumination super-resolution and 2-photon microscopy, patch clamp electrophysiology, and Western blots to explore the effects of the GS and RC LRRK2 mutations on striatal synapses. Both structural and functional changes are identified, and there are differences reported both between the types of LRRK2 mutations and between indirect and direct striatal pathways. More specifically, the authors observe increased expression and altered organization of glutamatergic AMPA receptors in LRRK2 mutants and an associated decreased in the frequency of EPSCs at these synapses. This was accompanied by changes in the nano-architecture of dendritic spines and ionotropic currents activated with 2-photon glutamate uncaging. In general, the reviewers felt that the study nicely integrates several approaches to evaluate and compare how indirect and direct striatal output pathway synapses are altered in 2 relevant LRRK2 mutations. The attention to both functional and structural changes within these pathways represents a significant experimental advance in our understanding of the implications of LRKK2 and its role in heritable Parkinson's disease.

Revisions:

1) Results, paragraph one. This information regarding Rab8A isn't very clear. Has the Rab8A protein been identified in SPNs previously? If not, then its identification should be reported first, followed by the description of experiments showing its phosphorylation in the mutant mice.

2) Results, subsection “Pathway specific functional alterations in the synapses of R1441C and G2019S SPNs”. The sentence describing differences in mEPSC frequency for the mutant mice in the dSPN pathway should be reworded for clarity. In this instance, the frequency was lower than the control genotype. However, the authors are using the word "decreased" as a noun, indicating that a change was observed within an experiment, which is clearly not the case.

3) In the same subsection the authors should indicate that no changes in "mean mEPSC" amplitudes were noted because they report below that changes were observed in subsets of these mEPSCs.

4) Results, subsection “Pathway specific functional alterations in the synapses of R1441C and G2019S SPNs”, last sentences. The authors report that whereas no changes in averaged mEPSC amplitudes were noted across genotypes, there were changes in subsets of these events that could be discerned using cumulative amplitude distributions. A satisfactory exploration of this is central to the work because of the complex relationship between GluA1 (increased) and the changes in mEPSC frequency (decreased) and amplitude (maybe an increase?) reported by the authors. Currently, the change in mEPSC amplitude is not clear from the way these data are presented in Figure 4, and the statistical evaluation of these data (K-S- test with Bonferroni post-hoc) does not permit an evaluation of where in the distribution of amplitudes these changes occur. The changes in amplitude distribution would best be evaluated by re-plotting the data using non-cumulative amplitude histograms for individual cells (or groups), and by estimating mean mEPSC amplitude modes using either Gaussian, or other statistical curves that fit the data more appropriately. Non-cumulative histograms should then be shown to demonstrate the differences in mEPSC sub-sets, and the number of cells showing a significant shift toward larger amplitude mEPSCs should be reported.

5) The statistical handling of the data in Figures 1F, 1G, and Figure 1—figure supplement 1C is inappropriate. Normalizing to the mean of the control group and getting rid of the error bar in that group negates a potentially important source of error. Use of a non-parametric analysis does not mitigate this concern. These data should be reanalyzed and re-graphed with the actual ratio values, which are already normalized to PSD95 or Homer1 expression. Sample sizes here were low, therefore this could require running additional samples if certain results drop out of significance upon re-analysis.

6) Currently, the discussion is very LRRK2-centric and doesn't venture as it should outside of models with this gene to larger questions concerning PD and basal ganglia circuitry, therefore it's not entirely clear where this work fits in with the larger field. Conspicuously absent is any link between this work and what is known about the direct and indirect pathway SPNs in PD. The discussion would therefore benefit from information previously published with regard to PD in humans or in animal models integrated with the present work.

[Editors' note: further revisions were suggested prior to acceptance, as described below.]

Thank you for resubmitting your article "Pathway-specific dysregulation of striatal excitatory synapses by LRRK2 mutations" for consideration by *eLife*.

Summary:

In this revision the authors have made several changes in-line with comments from the earlier review and this has improved the manuscript. However, an issue remains with the implementation of curvilinear fits to the mEPSC data that should be addressed.

Revisions:

1) Figure 4 and Figure 4—figure supplement 1 and Materials and methods: A description of how the Gaussian fits were obtained as well as the statistical analysis that was used to evaluate them should be included in the Materials and methods.

2) Figure 4—figure supplement 1: "Bin Center" is not an adequate label. This should be re-labeled either "Amplitude (pA)" or "Bin center (pA). Additionally, the pertinent data for these fits (mean amplitude, S.D. or S.E.M.) and goodness of fit information should be provided.

3) Figure 4—figure supplement 1: Why are the complete gaussian curves not shown? It is the peak of the inverted U-shaped curve that defines the mean amplitude of the responses falling into that bin. This is not easily observed without the full curve. Also, the means and errors (SEM or SD) of the amplitudes defined by the Gaussian curves should be given.

---

## [Author Response]

Revisions:1) Results, paragraph one. This information regarding Rab8A isn't very clear. Has the Rab8A protein been identified in SPNs previously? If not, then its identification should be reported first, followed by the description of experiments showing its phosphorylation in the mutant mice.

To highlight the increase in LRRK2 kinase function in the RC genotype mice we now have examined another LRRK2 substrate, a protein Rab10 that is part of the same family as Rab8A and is reported to be expressed in the striatum (e.g., Allen Brain Atlas, Dropviz).

We have also added an acknowledgement that Rab8A has not been extensively characterized in the striatum; however, low expression of Rab8A RNA has been reported in single cell sources like Dropviz, and our data show moderate expression of a band likely corresponding to p-Rab8A/total Rab8A (which is also increased in the RC genotype). Therefore, with the newly added data for Rab10 phosphorylation showing a similar trend to p-Rab8A, we have decided to keep both LRRK2 targets in this figure. Results paragraph one and Figure 1D.

2) Results, subsection “Pathway specific functional alterations in the synapses of R1441C and G2019S SPNs”. The sentence describing differences in mEPSC frequency for the mutant mice in the dSPN pathway should be reworded for clarity. In this instance, the frequency was lower than the control genotype. However, the authors are using the word "decreased" as a noun, indicating that a change was observed within an experiment, which is clearly not the case.

Reworded for clarity in the sentence noted by the reviewer, as well as in the following sentences.

3) In the same subsection the authors should indicate that no changes in "mean mEPSC" amplitudes were noted because they report below that changes were observed in subsets of these mEPSCs.

Reworded as requested.

4) Results, subsection “Pathway specific functional alterations in the synapses of R1441C and G2019S SPNs”, last sentences. The authors report that whereas no changes in averaged mEPSC amplitudes were noted across genotypes, there were changes in subsets of these events that could be discerned using cumulative amplitude distributions. A satisfactory exploration of this is central to the work because of the complex relationship between GluA1 (increased) and the changes in mEPSC frequency (decreased) and amplitude (maybe an increase?) reported by the authors. Currently, the change in mEPSC amplitude is not clear from the way these data are presented in Figure 4, and the statistical evaluation of these data (K-S- test with Bonferroni post-hoc) does not permit an evaluation of where in the distribution of amplitudes these changes occur. The changes in amplitude distribution would best be evaluated by re-plotting the data using non-cumulative amplitude histograms for individual cells (or groups), and by estimating mean mEPSC amplitude modes using either Gaussian, or other statistical curves that fit the data more appropriately. Non-cumulative histograms should then be shown to demonstrate the differences in mEPSC sub-sets, and the number of cells showing a significant shift toward larger amplitude mEPSCs should be reported.

We have carried out the analyses for mEPSC amplitudes as suggested by the reviewer. We added the plots of non-cumulative histograms of response amplitudes, and kept the cumulative distributions as a small inset, for those familiar with this way of representing the data. K-S tests were removed for both amplitude and frequency analyses. We have compared Gaussian fits to the amplitude histograms across cell class and genotype, reporting the results in the main text and new Figure 4—figure supplement 1 showing the fits. These analyses confirm the shift towards larger events selectively in the GFP-negative SPNs of the RC genotype. Subsection “Pathway specific functional alterations in the synapses of R1441C and G2019S SPNs”, (Figure 4D, Figure 4—figure supplement 1).

5) The statistical handling of the data in Figures 1F, 1G, and S1C is inappropriate. Normalizing to the mean of the control group and getting rid of the error bar in that group negates a potentially important source of error. Use of a non-parametric analysis does not mitigate this concern. These data should be reanalyzed and re-graphed with the actual ratio values, which are already normalized to PSD95 or Homer1 expression. Sample sizes here were low, therefore this could require running additional samples if certain results drop out of significance upon re-analysis.

Based on the reviewer’s recommendation, we ran two additional samples (for n=6), and we have graphed the data with the actual values normalized to PSD95 expression. The conclusions from this re-analysis remain the same. In addition, we have re-plotted the data of the PSD95/Homer normalization showing their values rather than the normalized mean of the control group. Results paragraph one, Figure 1F, G and Figure 1—figure supplement 1B, C.

6) Currently, the discussion is very LRRK2-centric and doesn't venture as it should outside of models with this gene to larger questions concerning PD and basal ganglia circuitry, therefore it's not entirely clear where this work fits in with the larger field. Conspicuously absent is any link between this work and what is known about the direct and indirect pathway SPNs in PD. The Discussion would therefore benefit from information previously published with regard to PD in humans or in animal models integrated with the present work.

We have added a new paragraph at the end of the Discussion where we place the data in a broader context.

[Editors' note: further revisions were suggested prior to acceptance, as described below.]

Revisions:1) Figure 4 and Figure 4—figure supplement 1 and Materials and methods: A description of how the Gaussian fits were obtained as well as the statistical analysis that was used to evaluate them should be included in the Materials and methods.

We fit the data with a gamma distribution, a non-negative asymmetrical distribution, because the mEPSC amplitude data were not normally or lognormally distributed (failing D'Agostino and Pearson, Shapiro-Wilk, and KS normality tests, p-values < 0.0001, alpha=0.05, for both raw and log-transformed data). Skew and kurtosis across the 6 datasets were 2.19±0.16 and 6.34±0.1, respectively, appropriate the gamma distribution.

For further comparison, continuous Gaussian and gamma distributions were fitted to the EPSC amplitude data by genotype and cell type using fitdist function in fitdistrplus library in R with the maximum likelihood estimation method. The Akaike information criterion (AIC) values were reported as goodness-of-fit criteria. The lower AIC values suggested a more appropriate fit using gamma distribution, which was overlayed in the histogram with bin width=1pA (Figure 4—figure supplement 1A).

**Author response table 1. resptable1:** 

Genotype/ Cell-type	Gaussian/ Normal AIC	Gamma AIC
*BAC, GFP-*	13166.75	11812.54
*RC, GFP-*	8252.910	7721.240
*GS, GFP-*	5373.360	4919.803
*BAC, GFP+*	10312.09	9476.535
*RC.GFP+*	7324.117	6825.080
*GS, GFP+*	5305.298	4880.877

To compare whether each distribution is statistically significantly different, we estimated the means of mEPSC amplitudes across different genotypes for GFP- and GFP+ neurons types using generalized linear modeling (GLM) with a gamma family distribution. All coefficients were statistically significant and were kept for posthoc pairwise comparison. To test whether mEPSC amplitudes differ across genotype by cell type, we used emmeans R package to summarize the estimated mean of EPSC amplitudes and applied Bonferroni correction at a confidence level alpha=0.05. Estimated responses (i.e., fitted mEPSC amplitude) were reported along with the corresponding lower and upper 95% confidence intervals (Figure 4—figure supplement 1B). Bonferroni corrected posthoc comparisons indicated that mEPSC amplitudes differ across genotype in GFP-, but not GFP+ neurons.

2) Figure 4—figure supplement 1: "Bin Center" is not an adequate label. This should be re-labeled either "Amplitude (pA)" or "Bin center (pA). Additionally, the pertinent data for these fits (mean amplitude, S.D. or S.E.M.) and goodness of fit information should be provided.

Figure 4—figure supplement 1 has been replaced, with posthoc amplitude mean comparisons and goodness of fit information provided in the legend, along with details in the Materials and methods section.

3) Figure 4—figure supplement 1: Why are the complete gaussian curves not shown? It is the peak of the inverted U-shaped curve that defines the mean amplitude of the responses falling into that bin. This is not easily observed without the full curve. Also, the means and errors (SEM or SD) of the amplitudes defined by the Gaussian curves should be given.

Updated curve fits are now shown (1) overlaid on the amplitude histogram for each genotype/cell type combination and (2) separately with the family of curves in Figure 4—figure supplement 1A. Amplitude estimates defined by GLM with gamma distribution are provided in Figure 4—figure supplement 1B.

The Author response table 2 summarized the means and errors (SE) of amplitudes, fitted by the GLM model:

**Author response table 2. resptable2:** 

Cell type	Genotype	Mean amplitude (pA)	SE	LCL*	UCL*
GFP-	*+/+ (BAC)*	9.33	0.106	9.08	9.59
	*+/GS*	8.26	0.138	7.95	8.61
	*+/RC*	10.11	0.143	9.78	10.47
GFP+	*+/+ (BAC)*	8.91	0.110	8.66	9.19
	*+/GS*	8.69	0.148	8.35	9.06
	*+/RC*	8.89	0.128	8.60	9.21

*LCL and UCL are 95% lower and upper confidence limits.

Bonferroni-corrected p values for posthoc comparisons between genotypes for each cell type:

**Author response table 3. resptable3:** 

Cell type	Comparison	Bonferroni corrected p-values
GFP-	*+/+ vs GS*	< 0.0001
	*+/+ vs RC*	< 0.0001
	*GS vs RC*	< 0.0001
GFP+	*+/+ vs GS*	0.7070
	*+/+ vs RC*	1.0000
	*GS vs RC*	0.9278

Besides the addition of a new Figure 4—figure supplement 1, the changes described above are now incorporated in the text (subsection “Pathway specific functional alterations in the synapses of R1441C and G2019S SPNs”), Materials and methods (Statistical analyses), and new Figure 4—figure supplement 1 legend.